# Bridging Theory and Practice in Link Representation with Graph Neural Networks

**Veronica Lachi**[*]
Fondazione Bruno Kessler
Trento, Italy
vlachi@fbk.eu

**Francesco Ferrini**[*]
University of Trento
Trento, Italy
francesco.ferrini@unitn.it

**Antonio Longa**
University of Trento
Trento, Italy
antonio.longa@unitn.it

**Bruno Lepri**
Fondazione Bruno Kessler
Trento, Italy
lepri@fbk.eu

**Andrea Passerini**
University of Trento
Trento, Italy
andrea.passerini@unitn.it

**Manfred Jaeger**
University of Aalborg
Aalborg, Denmark
jaeger@cs.aau.dk

## Abstract

Graph Neural Networks (GNNs) are widely used to compute representations of node pairs for downstream tasks such as link prediction. Yet, theoretical understanding of their expressive power has focused almost entirely on graph-level representations. In this work, we shift the focus to links and provide the first comprehensive study of GNN expressiveness in link representation. We introduce a unifying framework, the $k_\phi$-$k_\rho$-$m$ framework, that subsumes existing message-passing link models and enables formal expressiveness comparisons. Using this framework, we derive a hierarchy of state-of-the-art methods and offer theoretical tools to analyze future architectures. To complement our analysis, we propose a synthetic evaluation protocol comprising the first benchmark specifically designed to assess link-level expressiveness. Finally, we ask: does expressiveness matter in practice? We use a graph symmetry metric that quantifies the difficulty of distinguishing links and show that while expressive models may underperform on standard benchmarks, they significantly outperform simpler ones as symmetry increases, highlighting the need for dataset-aware model selection.

## 1 Introduction

Graph Neural Networks (GNNs) have achieved remarkable success across a wide range of tasks involving structured data, including node-level [19, 51, 36, 28], graph-level [18, 52], and link-level tasks [44, 58, 56, 13]. Despite the growing importance of link representation learning, the theoretical analysis of GNNs has so far focused almost exclusively on their expressiveness for graph-level representation [53, 37, 6]. In particular, it is well-established that message-passing GNNs are at most as powerful as the 1-dimensional Weisfeiler-Lehman (1-WL) test in distinguishing non-isomorphic graphs [53, 37]. However, their ability to distinguish *links* – that is, to generate discriminative representations of node pairs – remains far less understood.

It is well known that standard GNNs struggle to distinguish structurally different links, as they typically compute link embeddings by aggregating the representations of the two endpoint nodes. This node-centric strategy introduces a significant expressiveness bottleneck: links whose endpoints are automorphic may be mapped to identical representations, even if their structural roles in the graph differ [44, 58, 13]. To mitigate this issue, several methods have extended GNNs with structural

---

[*]Equal contribution

39th Conference on Neural Information Processing Systems (NeurIPS 2025).

features (SFs) [58, 56, 46, 13]. While such approaches are more expressive than standard GNNs, there is still no unifying framework to systematically characterize their discriminative power or organize them into a principled expressiveness hierarchy—unlike the case of graph-level representation, where the $k$-Weisfeiler-Lehman hierarchy provides a widely adopted standard [37, 53].

In this work, we bridge this gap by conducting a comprehensive investigation into the expressive power of GNN-based models for *link representation*. We observe that most existing methods for link representation combine two ingredients: a message-passing GNN for node encoding and a pairwise function for aggregating information about the target link. Building on this insight, we propose a *general theoretical framework* that captures a broad class of link representation methods. The framework characterizes models based on two key dimensions: the expressiveness of the functions involved and the radius of the neighborhood around the link that they access. It subsumes many state-of-the-art models and allows us to formally reason about their ability to distinguish links.

We explore link-level expressiveness along three axes. **First**, grounded in the proposed framework, we develop a theoretical foundation to analyze and *compare the expressiveness of link representation models*. Specifically, we derive criteria to assess a model's capacity to discriminate between different link structures, enabling us to organize existing methods into an expressiveness hierarchy and to position novel models within it. **Second**, we introduce a *synthetic evaluation protocol* designed to probe the expressiveness of link representation methods. This protocol comprises both a synthetic dataset and an accompanying evaluation procedure. While previous work has introduced synthetic tools for assessing graph-level expressiveness [7, 48, 1, 40, 3, 8], to the best of our knowledge, no analogous setup exists for links. Our procedure enables systematic, reproducible, and scalable comparisons across models. **Third**, we investigate the *practical relevance* of expressiveness on link representation by studying whether more expressive models lead to better performance in real-world scenarios. We propose a graph symmetry metric to quantify structural ambiguity among links, and we empirically show that while simple models suffice in low-symmetry settings, more expressive models significantly outperform them as symmetry increases.

## 2 Preliminaries

**Definition 2.1** (*graph*). A **graph** is a tuple $G = (V_G, E_G)$ where $V_G = \{1, \ldots, n\}$ is a set of nodes, $E_G \subseteq V_G \times V_G$ is a set of edges. To each graph is associated a node features matrix $\mathbf{X}^0 \in \mathbb{R}^{n \times f}$ and an adjacency matrix $\mathbf{A}_G \in \{0, 1\}^{n \times n}$ with $\mathbf{A}_{G_{u,v}} = 1$ if and only if $(u, v) \in E_G$. In our analysis, we consider simple, finite and undirected graphs. We will call $\mathcal{G}$ the set of all graphs.

**Definition 2.2** (*neighborhood*). Let $G = (V_G, E_G)$ be a graph. Given two nodes $u, v \in V_G$, $u$ is said to be $m$-far from $v$ if there exists a path of $m$ edges connecting $u$ and $v$. Given a node $v \in V_G$, we denote by $N^m(v)$ the set of nodes $m$-far away from $v$. In particular, the 1-hop neighborhood of $v$, denoted $N(v)$, corresponds to $N^1(v)$ and includes all nodes connected to $v$, i.e., $N(v) = \{u \in V_G \mid (v, u) \in E_G\}$. Finally, given two nodes $u, v \in V_G$, their joint $m$-hop neighborhood is defined as $N^m(u, v) = N^m(u) \cup N^m(v)$.

**Definition 2.3** (*link representation model*). A **link representation model** $M$ is a class of functions $F$ mapping node pairs in graphs to vector representations:

$$F : ((u, v), G, \mathbf{X}^0) \mapsto \mathbf{x}_{(u,v)} \in \mathbb{R}^d \tag{1}$$

with $G \in \mathcal{G}$, $\mathbf{X}^0 \in \mathbb{R}^{n \times f}$ node feature matrix and $u, v \in V_G$.

In this work, we adopt a broad notion of *link*, referring to any pair of nodes $(u, v) \in V_G \times V_G$, regardless of whether an edge between them actually exists in $G$, i.e., whether $(u, v) \in E_G$. Accordingly, the link representation model computes a vector representation for arbitrary node pairs, not only for existing edges. Throughout the paper, the term *link* will always refer to a generic node pair.

Models that compute link representations can be applied to a variety of downstream tasks. For instance, they can be used for *link prediction* [57, 35, 60], where the goal is to predict the existence of an edge between two nodes, for *link classification* [43, 47, 15], where different types of relationships are inferred and for *link regression* [33, 17], where real-valued properties associated with node pairs are estimated. In this work, we focus on message-passing (MP) link representation models, which have demonstrated superior performance and robustness across a wide range of benchmarks [32].

**Definition 2.4** (*GNN*). Let $G = (V_G, E_G)$ be a graph with feature matrix $\mathbf{X}^0$. A **GNN** iteratively updates the representation of nodes $v \in V_G$ following the propagation scheme:

$$\mathbf{x}_v^0 = \mathbf{X}_{[v,:]}^0 \qquad \mathbf{x}_v^l = \text{UPDATE}\left(\mathbf{x}_v^{l-1}, \text{AGGREGATE}\left(\{\mathbf{x}_u^{l-1} \mid u \in N(v)\}\right)\right). \tag{2}$$

We denote the final representation at the $L$-th layer as $\mathbf{x}_v^L = \rho(v, G, \mathbf{X}^0)$.

In the rest of this paper, whenever we refer to MP models, we refer to models that employ the iterative mechanism defined in Definition 2.4 to compute node representations.

A key theoretical question concerns GNNs expressive power, their ability to distinguish non-isomorphic inputs. Standard GNNs have been shown to be at most as powerful as the 1-WL test [37, 54], a heuristic for graph isomorphism based on iterative multiset aggregation [50]. To go beyond this limit, higher-order GNNs [37] aggregate over $k$-tuples of nodes, reaching the expressivity of the $k$-WL test; we denote by $k_\rho$ the smallest such $k$ such that $\rho$ is as powerful as $k$-WL, i.e, $k_\rho$ expresses the representational power of $\rho$ within the $k$-WL paradigm. In contrast to graph-level tasks, the expressiveness of message-passing models for link representation remains less explored [58, 44, 24]. We now provide formal definitions on what does it mean for two links to be different in a graph.

**Definition 2.5** (*node permutation*). A **node permutation** $\pi : \{1, \ldots, n\} \rightarrow \{1, \ldots, n\}$ is a bijective function that assigns a new index to each node of the graph. All the $n!$ possible node permutations constitute the permutation group $\Pi_n$. Given a subset of nodes $S \subseteq V_G$, we define the permutation $\pi$ on $S$ as $\pi(S) := \{\pi(i) | i \in S\}$. Additionally, we define $\pi(\mathbf{A}_G)$ as the matrix $\mathbf{A}_G$ with rows and columns permutated based on $\pi$, i.e., $\pi(\mathbf{A}_G)_{\pi(i), \pi(j)} = (\mathbf{A}_G)_{i,j}$.

**Definition 2.6** (*automorphism*). An **automorphism** on the graph $G = (V_G, E_G)$ is a permutation $\sigma \in \Pi_n$ such that $\sigma(\mathbf{A}_G) = \mathbf{A}_G$. All the possible automorphisms on a graph $G$ constitute the automorphism group $\Sigma_n^G$.

**Definition 2.7** (*automorphic links*). Let $G = (V_G, E_G)$ be a graph and $\Sigma_n^G$ its automorphism group. Two pairs of nodes $(u, v), (u', v') \in V_G \times V_G$ are said to be **automorphic links** ( $(u,v) \simeq (u', v')$ ) if exists $\sigma \in \Sigma_n^G$ such that $\sigma(\{u, v\}) = \{u', v'\}$.

The notion of automorphic links provides the foundation for evaluating models expressivity, as an expressive model should be able to distinguish between non automorphic links.

**Definition 2.8** (*more expressive*). Let $M_1$ and $M_2$ be two link representation models (Def. 2.3). $M_2$ is **more expressive** than $M_1$ ($M_1 \leq M_2$) if, for any graph $G = (V_G, E_G)$ and any pair $(u, v), (u', v') \in V_G \times V_G$ with $(u, v) \not\simeq (u', v')$:

$$\exists F_1 \in M_1 : F_1((u,v), G, \mathbf{X}^0) \neq F_1((u', v'), G, \mathbf{X}^0) \Rightarrow \exists F_2 \in M_2 : F_2((u,v), G, \mathbf{X}^0) \neq F_2((u', v'), G, \mathbf{X}^0).$$

## 3 The Expressive Power of MP-based Link Representation Methods

We propose a general framework for message-passing models that compute link representations, encompassing a broad class of existing methods as specific instances. By formalizing these models under a unified perspective, we enable a principled analysis of their expressive power. Leveraging this framework, we further establish an expressiveness-based hierarchy of existing approaches.

### 3.1 MP-based methods for link representation

Before defining the framework, we first review representative MP-based models, including Pure GNNs [30, 29, 45, 22, 53], NCN [46], NCNC [46], ELPH [13], BUDDY [13], Neo-GNN [56] and SEAL [58].

**Pure GNN Methods.** Pure GNN Methods learn representation $\mathbf{x}_{(u,v)} \in \mathbb{R}^d$ for each pair of node $(u, v)$ with $u, v \in V_G$ as $\mathbf{x}_{(u,v)} = g(\mathbf{x}_u^L, \mathbf{x}_v^L)$, where $g$ is an aggregation function and $\mathbf{x}_u^L, \mathbf{x}_v^L$ are the node representation of $u$ and $v$ learned by the GNN at the final layer $L$. In principle, any type of GNN can be used, and the function $g$ can be modeled using an MLP over any aggregation function over the feature vectors. In practice, the most commonly used pure GNN model is GAE [29].

**Neo-GNN.** Given a graph $G = (V_G, E_G)$, Neo-GNN computes a node representation matrix $\mathbf{Z}$ as:

$$\mathbf{Z} = \sum_{l=1}^{L} \beta^{l-1} \mathbf{A}_G^l \mathbf{X}^{\text{struct}} \quad \text{with} \quad \mathbf{X}^{\text{struct}} = \text{diag}(\mathbf{x}^{\text{struct}}), \; \mathbf{x}_v^{\text{struct}} = \mathcal{F}_\Theta((\mathbf{A}_G)_v) \tag{3}$$

where $\mathcal{F}_\Theta$ is a learnable function over the adjacency matrix $A_G$, $L$ is the maximum hop considered, and $\beta \in (0, 1)$ controls the weight of distant neighbors. The final link representation is computed as an aggregation of the node representation obtained through standard GNN and the one in $\mathbf{Z}$, namely $\mathbf{x}_{(u,v)} = g((\mathbf{z}_u, \mathbf{z}_v), (\mathbf{x}_u^L, \mathbf{x}_v^L))$.

**NCN.** NCN computes the representation of node pairs using the representations of their common neighbors. NCN defines the link representation as:

$$\mathbf{x}_{(u,v)} = \mathbf{x}_u^L \odot \mathbf{x}_v^L \parallel \sum_{i \in N(u) \cap N(v)} \mathbf{x}_i^L, \tag{4}$$

where $\mathbf{x}_u^L$ is the node representation at the final $L$−th layer of a GNN. NCNC extends NCN to deal with settings with incomplete topological information (e.g., for link prediction or graph completion), by considering not only nodes in $N(u) \cap N(v)$ but all nodes in $N(u) \cup N(v)$ and calculating for them the probability of being common neighbors with NCN.

**SEAL.** Given a graph $G = (V_G, E_G)$, for each link $(u, v)$, SEAL constructs an $h$-hop enclosing subgraph $G_{uv}^{(h)}$, which contains all nodes within $h$ hops of $u$ and $v$. Nodes in $G_{uv}^{(h)}$ are labeled using the Double-Radius Node Labeling (DRNL)[58], which assigns different labels based on distances to $u$ and $v$. The representation of link $(u, v)$ is obtained processing the graph $G_{uv}^{(h)}$ with a GNN.

**ELPH.** In ELPH, two types of count-based features are computed: (i) $A_{uv}[d_u, d_v]$, i.e., the number of nodes at distance $d_u$ from node $u$ and $d_v$ from node $v$; and (ii) $B_{uv}[d]$, the number of nodes at distance $d$ from $u$ and more distant than $d$ from $v$. These counts are estimated using *MinHash* (for Jaccard similarity) and *HyperLogLog* (for set cardinality). The link representation is obtained as:

$$\mathbf{x}_{(u,v)} = g(\mathbf{x}_u^L, \mathbf{x}_v^L, \; \{A_{uv}[d_u, d_v], B_{uv}[d] \mid d, d_u, d_v < [L]\}), \tag{5}$$

where $g$ in an aggregation function and $L$ is the final layer. To overcome ELPH's memory limitations, BUDDY precomputes structural features via offline graph traversals and sketching.

### 3.2 A General Framework for MP-based Link Representation Methods

Building on the definitions introduced in Section 3.1, we observe that MP-based approaches for link representation generally follow a common paradigm: they combine the representations of the two endpoint nodes with additional pairwise information extracted from their local neighborhoods, except for pure GNN methods, which rely solely on node representations. Although specific mechanisms and architectural designs vary, the core structure of these models can be systematically characterized by two key factors: (i) the link neighborhood radius, which defines the portion of the graph considered when extracting pairwise information for the link, and (ii) the expressiveness of the message-passing (MP) functions used to compute the node representations. Importantly, the link neighborhood radius

Table 1: Model formulations from Section 3.1 expressed within the $k_\phi$-$k_\rho$-$m$ framework. A '/' indicates that the corresponding component is not included in the model.

| Model | COMB | $g$ | $k_\phi$ | AGG | $\psi$ | $k_\rho$ | $m$ | $h$ |
|---|---|---|---|---|---|---|---|---|
| Pure GNN | / | $\odot$ | 1-WL | / | / | / | / | / |
| NCN | $\parallel$ | $\odot$ | 1-WL | $\sum$ | $\rho(i; G, \mathbf{X}^0)$ | 1-WL | 1 | $\mathbf{X}^0$ |
| ELPH | $\parallel$ | $\parallel$ | 1-WL | $\sum$ | $\rho(i; G, \mathbf{X}^1) \cdot \prod_{r=1}^{m} \prod_{d=1}^{m} \mathbb{1}_{dr}(i)$ | 1-WL | m | $\mathbf{x}_i^1 = 1$ |
| Neo-GNN | $\parallel$ | $\parallel$ | 1-WL | $\sum$ | $b \cdot \rho(i; G, \mathbf{X}^0)$ with $b = \sum_{r=1}^{m} \sum_{d=1}^{m} A_{uv}^r \cdot A_{uv}^d$ | 1-WL | m | $\mathbf{X}^0$ |
| SEAL | / | / | / | $\sum$ | $\rho(i; G, \mathbf{X}^D)$ | 1-$|N^m(u,v)|$-WL | m | $\mathbf{x}_i^D = \mathbf{x}_i^0 \parallel \min_{u,v}(\delta(i, u), \delta(i, v)) + 1$ |

is distinct from the radius induced by the depth of the MP function itself. Motivated by these common factors, we introduce a general framework that unifies existing approaches under a common formalism. In this framework, the link representation is constructed by combining: (1) the representations of the two endpoint nodes, and (2) a representation of the link's $m$-order neighborhood, each computed via MP functions that may have different levels of expressive power and different number of layers.

We formalize this in the following definition, which we refer to as the $k_\phi$-$k_\rho$-$m$ *framework* for link representation. For ease of readability, we omit the graph $G \in \mathcal{G}$ as an explicit argument in message-passing and link-representation functions, assuming it is implicitly defined as the underlying structure on which these functions operate.

**Definition 3.1** ($k_\phi$-$k_\rho$-$m$ *framework* )**.** Given a graph $G = (V_G, E_G)$ with feature matrix $\mathbf{X}^0$, a MP link representation model $M$ belongs to the $k_\phi$-$k_\rho$-$m$ framework if its functions can be expressed as:

$$F((u,v), \mathbf{X}^0) = \text{COMB}\Big(g\big(\phi(u, \mathbf{X}^0), \phi(v, \mathbf{X}^0)\big), \text{AGG}\big(\{f(i, u, v, \mathbf{X}^0) \mid i \in \bigcup_{j=0}^{m} N^j(u,v)\}\big)\Big) \quad (6)$$

$$f(i, u, v, \mathbf{X}^0) = \psi\big(\rho(i, h(u, v, \mathbf{X}^0)), u, v\big) \quad (7)$$

where $\phi$ and $\rho$ are MP functions (Definition 2.4) with expressive power respectively of $k_\phi$ and $k_\rho$, $h(u, v, G, \mathbf{X}^0) \in \mathbb{R}^{n \times d}$ is a new node feature matrix computed using the original feature matrix and possibly pairwise information, $\psi$ scales the message passing representations by a coefficient incorporating pairwise information from the graph, AGG is an aggregation function over the representations of nodes in $\bigcup_{j=0}^{m} N^j(u,v)$ and COMB combine the endpoints representations with the neighborhood representation.

Table 1 provides the formulation of all the models presented in Section 3.1 using the $k_\phi$-$k_\rho$-$m$ framework, while Figure 1 provides a visualization of the framework components. Through the $k_\phi$-$k_\rho$-$m$ framework, we can systematically study the expressive power of MP-based link representation methods. Let $\mathcal{M}$ be the set of link representation models expressible by the $k_\phi$-$k_\rho$-$m$ framework. Let $M \in \mathcal{M}$, we indicate with $k_\phi^M$-$k_\rho^M$-$m^M$ the specific value that $k_\phi$-$k_\rho$-$m$ assume in the model $M$.

**Theorem 3.2.** *Let $M \in \mathcal{M}$, the following hold:*

1. *If $m^M = 0$, then, regardless of $k_\phi^M$, $M$ is not able to distinguish between links whose endpoints are automorphic, i.e.,*

$$\forall F \in M, \forall (u,v), (u',v') \text{ s.t. } \exists \sigma_1, \sigma_2 \in \Sigma_n^G \text{ with } \sigma_1(u) = u' \wedge \sigma_2(v) = v' \quad (8)$$

$$F((u',v'), \mathbf{X}^0) = F((u,v), \mathbf{X}^0) \quad (9)$$

   *therefore, automorphic links whose endpoints are automorphic will be assigned to the same representation. Moreover, given $M_1, M_2 \in \mathcal{M}$, if $m^{M_1} = 0$, while $m^{M_2} > 0$ and COMB is injective, regardless of $k_\phi^{M_1}$ and $k_\phi^{M_2}$ , $M_1 \leq M_2$.*

2. *Let $M_1, M_2 \in \mathcal{M}$ with $k_\rho^{M_1} = k_\rho^{M_2} = $ 1-WL and let $l^{M_1}, l^{M_2}$ be the number of layers used respectively by $M_1, M_2$. If $m^{M_1} + l^{M_1} \leq m^{M_2} + l^{M_2}$, then $M_1 \leq M_2$.*

3. *Let $M_1, M_2 \in \mathcal{M}$ with $m^{M_1} = m^{M_2}$. If $k_\rho^{M_1} \leq k_\rho^{M_2}$, then $M_1 \leq M_2$.*

Figure 1: $k_\phi$-$k_\rho$-$m$ framework: nodes representations in the m-order neighborhood of target link are calculated using a MP function $\rho$ with possibly modified node features; these representations are aggregated and combined with representations of endpoints of target link obtained through MP function $\phi$.

Proof is provided in Appendix A. Intuitively, the theorem highlights that relying solely on the representations of the endpoints through $g$ and $\phi$ is fundamentally limited: even if $\phi$ is highly expressive, such models fail to distinguish links whose endpoints are automorphic. This limitation arises from the fact that the $k$-WL algorithm, preserves graph automorphisms for every $k$ [34, 16, 11]. Instead, a more efficient and expressive modeling strategy is to compute $\rho$ over a suitable $m$-order neighborhood using an appropriate number of MP layers. In many models, the number of such layers is a design choice: for instance, *ELPH* encodes link information by counting nodes at various distances from the link, which effectively corresponds to zero message-passing layers. Conversely, in *Neo-GNN*, node representations are computed through a function on the adjacency matrix, effectively implementing a single round of message passing. This distinction is crucial: even models with large neighborhoods $m$ but few message-passing layers can be outperformed—both in expressiveness and efficiency—by models with smaller $m$ but deeper architectures. Thus, controlling expressiveness via the number of layers offers a more principled and computationally efficient path. Finally, increasing the expressiveness of the base model $\rho$ on the $m$-order neighborhood leads to more expressive representations. One example of model with more expressive $\rho$ is SEAL which increases expressiveness by adding link-aware positional encodings to node features, effectively simulating the power of 1-$|N^m(u,v)|$-WL, which is strictly more expressive than 1-WL [59]. However, increasing expressivity beyond 1-WL comes at a computational cost: for example, the complexity of $k$-WL is $O(n^{k+1} \log n)$ [25]. Details on the computational costs of the models are provided in Appendix D.

Using Theorem 3.2 it is possible to study the expressiveness of existing methods. The following Theorem introduces a hierarchy of existing methods.

**Theorem 3.3.** *The following hold:*

1. *For any radius and number of layers, NCN, Neo-GNN, ELPH, SEAL are more expressive than Pure GNNs;*

2. *if $l^{NCN} \geq m^{Neo-GNN}$, NCN is more expressive then Neo-GNN;*

3. *if $m^{Neo-GNN} \geq m^{ELPH} - 1$, Neo-GNN is more expressive than ELPH;*

4. *if $l^{NCN} \geq m^{ELPH} - 1$, NCN is more expressive then ELPH;*

5. *SEAL is more expressive then NCNC, NCN, ELPH, Neo-GNN.*

Proof is provided in Appendix A and rely on Theorem 3.2 and the formal expression of models within the theoretical framework described in Table 1.

The $k_\phi$-$k_\rho$-$m$ framework serves as a foundation for analyzing the expressiveness of link representation models in a principled and structured way. In the next section, we complement this theoretical perspective by introducing a synthetic evaluation protocol designed to empirically assess the expressiveness of link representation models.

## 4   A Synthetic Evaluation Procedure for Link Representation Expressiveness

We introduce a synthetic protocol to empirically assess a MP model's ability to assign distinct representations to structurally different links. The protocol is designed to help evaluate new link representation models in a simple and controlled setting. It consists of a synthetic dataset and an evaluation framework. The synthetic dataset and the code are publicly available[2].

### 4.1   `LR-EXP`: A Synthetic Benchmark for Link-Level Expressiveness

While several synthetic datasets have been proposed to evaluate the expressive power of GNNs in distinguishing non-isomorphic graphs [7, 48, 1, 40, 3, 8], a comparable benchmark for assessing link-level expressiveness is still missing, despite its potential utility as a fast and controlled way to empirically assess the expressivity of both existing and new models. To address this gap, we propose a novel synthetic dataset, `LR-EXP`, aimed at measuring a model's ability to assign different outputs to non-automorphic links. To make the distinction task challenging, the dataset includes non-automorphic links whose endpoint nodes share the same 1-WL colors.

---

[2]`https://anonymous.4open.science/r/link-representation-gnn-8124/README.md`

This is a critical design choice: non-automorphic links with differently colored nodes are trivially distinguishable by any pure GNN with an injective aggregation function $g$. `LR-EXP` includes 1400 graphs, each containing at least one such pair of non-automorphic links. Each graph is generated in two steps: (1) sample an Erdős–Rényi graph with $n \in \{5, \ldots, 17\}$ nodes and edge probability $p \sim \mathcal{U}(0, 1)$; (2) duplicate the graph and add inter-block edges with probability $p' \sim \mathcal{U}(0, 1)$. This construction introduces rich symmetries and increases the probability of obtaining node pairs that are non-automorphic yet indistinguishable under the WL test. We retain only those graphs where such link pairs exist, and use them to define training, validation, and test sets. Figure 2 shows examples of these link pairs. Importantly, while `LR-EXP` is relatively small in scale, this is by design: our goal is not to evaluate scalability, but to isolate and probe expressivity. Since expressiveness concerns a model's ability to produce different outputs for structurally distinct inputs, rather than generalization over large data volumes, dataset size is not a confounding factor for the phenomenon we aim to study.

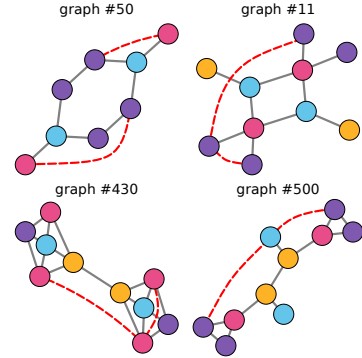

Figure 2: Examples of `LR-EXP` graphs. Node colors show WL colors; dashed lines mark test links that are non-automorphic but indistinguishable for standard GNNs.

## 4.2 Evaluation Framework

Our training and evaluation approach is inspired by the method proposed in Wang and Zhang [49]. In particular, we employ a siamese network design [9], with central component consisting of two identical models with identical parameters, each processing one link independently to generate separate embeddings. For a pair of links $(u, v) \not\simeq (u', v') \in V_G \times V_G$, and a model $F$, the model generates a corresponding pair of embeddings. The model is trained with the goal of encouraging separation of links embeddings using the contrastive loss:

$$L(F, (u, v), (u', v'), \mathbf{X}^0) = \max \left( 0, \frac{F((u, v), \mathbf{X}^0) \cdot F((u', v'), \mathbf{X}^0)}{\|F((u, v), \mathbf{X}^0)\|\|F((u', v'), \mathbf{X}^0)\|} \right), \qquad (10)$$

Contrastive loss functions are often defined with an explicit margin term which controls the degree of dissimilarity enforced between negative pairs. In our case, the goal is to fully minimize similarity between representations of non-automorphic links. Following the approach in [48], we deliberately adopted a zero-margin formulation to push the model toward maximal dissimilarity between such pairs. This choice aligns with our objective of measuring and testing expressiveness, where even small overlaps in representation between structurally different links would be undesirable.

To evaluate models, we compare their outputs on pairs of non-automorphic links. If the resulting representations differ sufficiently, this indicates successful discrimination. However, selecting an appropriate threshold is challenging: a high threshold may cause false negatives, where true distinctions fall below the threshold; a low threshold may cause false positives, where random noise creates apparent differences. To address this, we adopt the Reliable Paired Comparison method [49], which assesses groups of results using two components: the Major Procedure, quantifying representation differences, and the Reliability Check, measuring internal variability. A pair is considered distinguishable only if both checks are passed. As all pairs are non-automorphic, the task is framed as one-class classification, and we report precision as the fraction of correctly distinguished pairs. See Appendix B.1 for details.

## 4.3 Experimental results on `LR-EXP`

We evaluated the expressive power of all models from Section 3.1 on `LR-EXP`. As discussed in Section 3.2, some models treat $m$ as a fixed design choice while keeping the number of layers $l$ as a tunable parameter, and vice versa. Even in models where $m$ is left as a free parameter, practical implementations impose constraints due to computational costs. For example, both ELPH and SEAL limit $m$ to a maximum of 3. In our experiments, we adopt $m = 3$ for all models that allow it, ensuring a fair comparison. We set the number of layers $l = 3$ across all methods, which strikes a balance between expressiveness and avoiding oversmoothing effects [10]. Under this configuration, Theorem

3.3 shows that NCN is more expressive than ELPH and matches the expressiveness of Neo-GNN, while SEAL is the most expressive method.

Table 2: Test precision (mean $\pm$ std over 5 runs) for different models. Higher model expressiveness leads to higher test precision.

| Model | Test Precision(%) |
|-------|-------------------|
| GCN | 0 ($\pm$ 0) |
| GAT | 0 ($\pm$ 0) |
| GAE | 0 ($\pm$ 0) |
| SAGE | 0 ($\pm$ 0) |
| BUDDY | 45 ($\pm$ 1) |
| ELPH | 62 ($\pm$ 7) |
| Neo-GNN | 75 ($\pm$ 2) |
| NCN | 75 ($\pm$ 1) |
| SEAL | 97 ($\pm$ 0) |

The experimental results on LR-EXP provide empirical validation of our theoretical framework. Table 4.3 shows that Pure GNN methods (*GCN*, *GAT*, *GAE*, *SAGE*) completely fail to distinguish non-automorphic links, achieving zero precision. This is a consequence of their fundamental limitation - the automorphic node problem we analyzed in Section 3.2, which prevents them from generating distinct representations for links composed by automorphic nodes. The remaining methods demonstrate progressively better performance as their expressive power increases. The performance gap between *BUDDY* (45%) and *ELPH* (62%) stems from *BUDDY*'s approximate counting of common neighbors (Section 3.1), which reduces its expressiveness as shown in [13]. *Neo-GNN* and *NCN*[3] achieve comparable precision (75%), outperforming both *ELPH* (62%) and *BUDDY* (45%). Finally, *SEAL* demonstrates its high expressiveness, being able to correctly distinguish 97% of links. Further results as we vary the number of layers $l$ and the radius $m$ are in Appendix. Appendix B.2 reports robustness checks under variations of the significance level employed in the Reliable Paired Comparison.

## 5 Do More Expressive Models Perform Better in Real-World Benchmarks?

An interesting and practically relevant question is whether increased expressivity in link representation translates into better performance on downstream tasks involving real-world data [27, 38]. Among the various tasks enabled by expressive link representations, we focus on link prediction due to its broad impact across domains such as recommender systems [55], knowledge graph completion [41], and biological interaction prediction [26]. Intuitively, link prediction should particularly benefit from models with high expressive power when the dataset contains many non-automorphic links whose endpoint nodes are automorphic, a setting where Pure GNNs struggle. In this section, we aim to answer the following three questions: (1) How can we formally quantify the presence of non-automorphic links with automorphic endpoints in a dataset? (2) Are there real-world link prediction benchmarks where this property is especially prevalent? (3) How do models with varying levels of expressiveness perform across datasets with different levels of structural difficulty?

We address these questions by first introducing a metric to quantify the link-level symmetry of a dataset. We then apply this metric to a collection of real-world link prediction benchmarks and analyze their structural difficulty. Finally, we evaluate the performance of several state-of-the-art models with different expressive capabilities across these datasets. Code is available at[4].

### 5.1 A Metric for Measuring the Symmetry of Graphs

We introduce a metric to quantify the presence of non automorphic links with automorphic endpoints in a dataset. This occurs when the underlying graph exhibits a high number of non-trivial automorphisms that is, intuitively, when the graph is highly symmetric. The symmetry of a graph is tightly connected to the concept of *node orbits*, which we define below.

**Definition 5.1** (*orbit*). The **orbit** of a vertex $v \in V_G$ under the action of the automorphism group $\Sigma_n^G$ is defined as the set of nodes of $V_G$ to which $v$ can be mapped via an automorphism $\sigma \in \Sigma_n^G$. Formally, $\text{Orb}(v) = \{\sigma(v) \mid \sigma \in \Sigma_n^G\}$. We denote with $O_G = \{\text{Orb}(v) \mid v \in V_G\}$ the set of all the orbits of $V$ with respect to the action of $\Sigma_n^G$.

Following [4], it is possible to define a metric that mesures how much a graph is symmetric as follows:

---

[3]We do not report results for NCNC, as it coincides with NCN for fully observed topologies.

[4]https://anonymous.4open.science/r/link-representation-gnn-8124/README.md

Table 3: MRR on real-world link prediction datasets sorted by increasing $\hat{r}_G$. The top three results are highlighted using **first**, **second**, and **third**. OOM means Out Of Memory, >24h means more than 24 hours. As symmetry increases, only more expressive models maintain top performance. Results are averaged over 5 runs with different random seeds; standard deviations are reported in Appendix F.

| | | | | | | | Graph Symmetry ($\rightarrow$) | | | | | |
|---|---|---|---|---|---|---|---|---|---|---|---|---|
| **Models** | OGBL CITATION2 | CORA | CITESEER | PUBMED | OGBL DDI | OGBL PPA | OGBL COLLAB | NSC | YST | GRQ | AIFB | EDIT TSW |
| $\hat{r}_G$ | 0.01 | 0.02 | 0.02 | 0.08 | 0.10 | 0.12 | 0.13 | 0.16 | 0.23 | 0.30 | 0.41 | 0.67 |
| *GCN* | 19.98 | **16.61** | 21.09 | 7.13 | **13.46** | 26.94 | **6.09** | 26.79 | 1.17 | 6.89 | 10.85 | 4.12 |
| *GAT* | OOM | 13.84 | 19.58 | 4.95 | **12.92** | | 4.18 | 2.47 | 0.40 | 0.63 | 1.07 | 0.40 |
| *SAGE* | **22.05** | 14.74 | 21.09 | **9.40** | 12.60 | 27.27 | 5.53 | 20.32 | 1.77 | 1.34 | 0.54 | 8.35 |
| *GAE* | OOM | **18.32** | **25.25** | 5.27 | 3.49 | OOM | OOM | 16.61 | 2.32 | 2.32 | 10.98 | 3.54 |
| *BUDDY* | 19.17 | 13.71 | 22.84 | 7.56 | 12.43 | 27.70 | **5.67** | 18.40 | **13.68** | 46.23 | **13.73** | **22.66** |
| *Neo-GNN* | 16.12 | 13.95 | 17.34 | **7.74** | 10.86 | 21.68 | 5.23 | 22.86 | 5.78 | 29.51 | 13.11 | 8.10 |
| *NCN* | **23.35** | 14.66 | **28.65** | 5.84 | **12.86** | **35.06** | 5.09 | **30.36** | 11.99 | **48.45** | 6.47 | **10.32** |
| *NCNC* | 19.61 | **14.98** | **24.10** | **8.58** | >24h | **33.52** | 4.73 | **30.20** | **13.54** | **47.60** | OOM | 10.30 |
| *SEAL* | **20.60** | 10.67 | 13.16 | 5.88 | 9.99 | **29.71** | **6.43** | **30.85** | **17.51** | **56.72** | **16.30** | **25.82** |

**Definition 5.2** (*graph symmetry measure $r_G$*). Given a graph $G = (V_G, E_G)$, its **symmetry measure** is calculated as:

$$r_G = 1 - \frac{|O_G| - 1}{|V_G| - 1} \tag{11}$$

where $|O_G|$ denotes the number of orbits of the graph and $|V_G|$ is the number of nodes.

Computing all node orbits is computationally expensive. However, the number of orbits can be efficiently approximated using the number of distinct colors assigned by the WL test at convergence $WL_G$ [31, 39], i.e.,

$$\hat{r}_G = 1 - \frac{|WL_G| - 1}{|V_G| - 1} \tag{12}$$

The approximation in Equation 12 reflects how closely the WL-induced partition (used in Equation 11) matches the true orbit partition. Although there is no complete characterization of WL failures in distinguishing nodes from different orbits, it is said that WL gives different representations to *almost all* nodes that belong to different orbits [2]. A practical example is presented in [39], where the authors explicitly compare WL partitions and orbit partitions on the Alchemy dataset [14]. Out of 202,579 graphs, only in 4 graphs the partition of the nodes induced by the WL is different wrt the orbits, highlighting that the WL approximation is extremely close in practice.

## 5.2 Symmetric Real-World Datasets for Link Prediction

We selected twelve datasets for link prediction, including well-known citation networks such as CORA and CITESEER, large-scale benchmarks from the Open Graph Benchmark suite like OGBL-CITATION2 and OGBL-COLLAB, and structurally complex knowledge and ontology graphs such as AIFB, EDIT-TSW. More details about datasets can be found in Appendix E.1. For each dataset, we compute the proposed symmetry metric $\hat{r}_G$ (Equation 12). The computed values are reported in the first row of Table 3, with datasets sorted in ascending order of $\hat{r}_G$. As shown, the datasets exhibit widely varying degrees of symmetry. For instance, OGBL-CITATION2, CORA, and PUBMED display very low $\hat{r}_G$ values, largely due to the presence of informative node features that yield many unique WL colors. At the other end of the spectrum, datasets such as EDIT-TSW, which lacks node features and exhibit high topological regularity, show significantly higher symmetry scores.

Table 3 reports the results, in terms of the standard link prediction metric MRR, achieved by the models described in Section 3.1. The models are listed in ascending order of theoretical expressiveness, as characterized by Theorem 3.3. Since we are now evaluating under the standard link prediction setting (i.e., graphs with missing edges), we also include the NCNC variant of NCN. For ELPH, we report results for BUDDY, its more scalable version. All evaluations are conducted under the challenging and realistic HeaRT setting [32]. Further details on model architectures and hyperparameter selection are provided in Appendix E, while the complete results, including standard deviations and an additional evaluation metric, are reported in Appendix F.

The results reveal a clear trend: as $\hat{r}_G$ increases, more expressive models become necessary. In the five datasets with the lowest symmetry, pure GNN-based methods consistently rank among the top three performers. However, as symmetry increases, their performance deteriorates and none of them appears in the top three on the six most symmetric datasets. Conversely, SEAL, the most expressive model in our hierarchy, ranks first across all high-symmetry datasets. This provides a key practical insight: while simple, less expressive models may suffice when node features are informative and structural ambiguity is limited, datasets with high link symmetry demand more powerful architectures.

## 6   Conclusion

We propose a theoretical framework that formally compares a wide range of existing models and establishes a hierarchy of them. We introduced LR-EXP, the first synthetic benchmark explicitly designed to evaluate link-level expressiveness. Finally, we demonstrated that increased expressiveness translates into performance gains on real-world link prediction tasks, especially in structurally symmetric graphs. These findings highlight that model selection should account for dataset symmetry: while simple models may suffice in low-symmetry settings, high-expressiveness architectures are essential when structural ambiguity is present.

**Limitations and Future Work**   Our framework is tailored to MP models and does not directly account for recent advances in transformer-based architectures or spectral methods for link representation. Extending the framework to encompass these directions may provide additional insights, making this an interesting and valuable direction for future work. Furthermore, the high-symmetry datasets used in our evaluation are relatively small, highlighting the need for larger and more robust benchmarks to properly assess model generalization and advance research in this area [5].

## Acknowledgments

Funded by the European Union. Views and opinions expressed are however those of the author(s) only and do not necessarily reflect those of the European Union or the European Health and Digital Executive Agency (HaDEA). Neither the European Union nor the granting authority can be held responsible for them. Grant Agreement no. 101120763 - TANGO. We acknowledge the support of the MUR PNRR project FAIR - Future AI Research (PE00000013) funded by the NextGenerationEU. This work was also supported by Ministero delle Imprese e del Made in Italy (IPCEI Cloud DM 27 giugno 2022 – IPCEI-CL-0000007)

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

# A Proofs

***Proof of Theorem 3.2.*** We prove theorem by addressing each of its three components individually.

1. Let $M \in \mathcal{M}$. If $m^M = 0$, then, regardless of $k_\phi^M$, $M$ is not able to distinguish between links whose endpoints are automorphic, i.e.,

$$\forall F \in M, \forall (u,v), (u',v') \text{ s.t. } \exists \sigma_1, \sigma_2 \in \Sigma_n^G \text{ with } \sigma_1(u) = u' \wedge \sigma_2(v) = v' \qquad (13)$$

$$F((u',v'), \mathbf{X}^0) = F((u,v), \mathbf{X}^0) \qquad (14)$$

   therefore, automorphic links whose endpoints are automorphic will be assigned to the same representation. Moreover, given $M_1, M_2 \in \mathcal{M}$, if $m^{M_1} = 0$, while $m^{M_2} > 0$ and COMB is injective, regardless of $k_\phi^{M_1}$ and $k_\phi^{M_2}$, $M_1 \leq M_2$.

   ***Proof*** Suppose $m^M = 0$, i.e., $M$ is composed only of the functions $\phi$ and $g$, i.e., its functions are:

   $$F((u,v), \mathbf{X}^0) = g\big(\phi(u, \mathbf{X}^0), \phi(v, \mathbf{X}^0)\big).$$

   Assume $\phi$ is maximally expressive at the node level (note that no GNN can achieve this, as the $k$-WL algorithm, preserves graph automorphisms for every $k$ [34, 16, 11]), i.e., it assigns representations to nodes solely based on their automorphism classes. Then for any automorphisms $\sigma_1, \sigma_2 \in \Sigma_n^G$ such that $\sigma_1(u) = u'$ and $\sigma_2(v) = v'$, it follows that

   $$\phi(u, \mathbf{X}^0) = \phi(u', \mathbf{X}^0), \quad \phi(v, \mathbf{X}^0) = \phi(v', \mathbf{X}^0).$$

   Thus, regardless of the function $g$, we obtain

   $$g\big(\phi(u, \mathbf{X}^0), \phi(v, \mathbf{X}^0)\big) = g\big(\phi(u', \mathbf{X}^0), \phi(v', \mathbf{X}^0)\big),$$

   i.e., the model cannot distinguish between links whose endpoints are automorphic. If $(u,v)$ and $(u',v')$ are not themselves automorphic pairs, then the model fails to differentiate them, thus assigning the same representation to non-automorphic links.

   Now consider $M_1 \in \mathcal{M}$ with all functions $F \in M_1$ composed only of $\phi$ and $g$, while $M_2 \in \mathcal{M}$ with all functions $F \in M_2$ that also include a component $f$ (on the $m$-order neighborhood). Using Definition 2.8, we show that $M_2$ is more expressive.

   Let $F_1 \in M_1$, $F_2 \in M_2$, $(u,v), (u',v') \in V_G \times V_G$ such that $(u,v) \not\simeq (u',v')$ and

   $$F_1((u,v), \mathbf{X}^0) = F_1((u',v'), \mathbf{X}^0),$$

   but

   $$F_2((u,v), \mathbf{X}^0) \neq F_2((u',v'), \mathbf{X}^0).$$

   This implies that

   $$g\big(\phi(u, \mathbf{X}^0), \phi(v, \mathbf{X}^0)\big) = g\big(\phi(u', \mathbf{X}^0), \phi(v', \mathbf{X}^0)\big),$$

   but due to the presence of a component $f$ and the use of an injective COMB, we get

   $$F_2((u,v), \mathbf{X}^0) \neq F_2((u',v'), \mathbf{X}^0),$$

   demonstrating by Definition 2.8 that $M_2$ is more expressive than $M_1$.

2. Let $M_1, M_2 \in \mathcal{M}$ with $k_\rho^{M_1} = k_\rho^{M_2} = 1$-WL and let $l^{M_1}, l^{M_2}$ be the number of layers used respectively by $M_1, M_2$. If $m^{M_1} + l^{M_1} \leq m^{M_2} + l^{M_2}$, then $M_1 \leq M_2$.

   ***Proof*** Let $F_1 \in M_1$, $F_2 \in M_2$. Assume $l^{M_2} \geq l^{M_1} + m^{M_1} - m^{M_2}$ and $l^{M_1} \leq l^{M_2}$ and $m^{M_1} > m^{M_2}$.

   Let $(u,v), (u',v') \in V_G \times V_G$ such that

   $$F_1((u,v), G, \mathbf{X}^0) \neq F_1((u',v'), G, \mathbf{X}^0).$$

   Let $c_u^\ell$ denote the color assigned to node $u$ after $\ell$ iterations of the 1-WL algorithm. Assume $c_u^\ell = c_{u'}^\ell$ and $c_v^\ell = c_{v'}^\ell$; otherwise, $F_2$ alone could distinguish between $(u,v)$ and $(u',v')$, and the proof would be complete.

We divide the proof into three steps. Since $F_1$ distinguishes $(u, v)$ from $(u', v')$ while $c_u^\ell = c_{u'}^\ell$ and $c_v^\ell = c_{v'}^\ell$, this implies that:

$$\left\{ \rho_1(w, \mathbf{X}) \,\middle|\, w \in \bigcup_{j=0}^{m^{M_1}} N^j(u, v) \right\} \neq \left\{ \rho_1(i, \mathbf{X}) \,\middle|\, i \in \bigcup_{j=0}^{m^{M_1}} N^j(u', v') \right\} \tag{1}$$

Let's consider the smaller neighborhood $m^{M_2}$:

$$\left\{ \rho_1(w, \mathbf{X}) \,\middle|\, w \in \bigcup_{j=0}^{m^{M_2}} N^j(u, v) \right\} \quad \text{and} \quad \left\{ \rho_1(i, \mathbf{X}) \,\middle|\, i \in \bigcup_{j=0}^{m^{M_2}} N^j(u', v') \right\}.$$

If these sets are already different, then $F_2$ (with smaller neighborhood but same $\rho_1$) suffices, and the proof is complete. If instead they are equal, then from (1) it follows that the difference lies in the sets:

$$\left\{ \rho_1(w, \mathbf{X}) \,\middle|\, w \in \bigcup_{j=m_2+1}^{m^{M_1}} N^j(u, v) \right\} \neq \left\{ \rho_1(i, \mathbf{X}) \,\middle|\, i \in \bigcup_{j=m_2+1}^{m^{M_1}} N^j(u', v') \right\}.$$

Since $F_2$ uses a number of layers $l^{M_2} \geq l^{M_1} + m^{M_1} - m^{M_2}$, it can compensate for the smaller neighborhood radius $m_2$ by using more layers to reach the same depth of information as $F_1$. Thus, using $\rho_2$ in $F_2$, we obtain:

$$\left\{ \rho_2(w, \mathbf{X}) \,\middle|\, w \in \bigcup_{j=0}^{m^{M_2}} N^j(u, v) \right\} \neq \left\{ \rho_2(w, \mathbf{X}) \,\middle|\, w \in \bigcup_{j=0}^{m^{M_2}} N^j(u', v') \right\},$$

which implies

$$F_2((u, v), \mathbf{X}^0) \neq F_2((u', v'), \mathbf{X}^0),$$

and by Definition 2.8, this yields $M_1 \leq M_2$.

3. Let $M_1, M_2 \in \mathcal{M}$ with $m^{M_1} = m^{M_2}$. If $k_\rho^{M_1} \leq k_\rho^{M_2}$, then $M_1 \leq M_2$.

   ***Proof*** Let $m^{M_1} = m^{M_2} = m$ and $k_\rho^{M_1} \leq k_\rho^{M_2}$. Let $F_1 \in M_1$ and $F_2 \in M_2$ and $(u, v), (u', v') \in V_G \times V_G$ such that

$$F_1((u, v), G, \mathbf{X}^0) \neq F_1((u', v'), G, \mathbf{X}^0).$$

Assume that $c_u^\ell = c_{u'}^\ell$ and $c_v^\ell = c_{v'}^\ell$, i.e., $u$ and $u'$, as well as $v$ and $v'$, have the same WL color after $\ell$ iterations. Otherwise, $F_2$ could already distinguish the links based on $k_\rho^{M_2}$ alone, and the proof would be trivial.

Now, since $F_1$ can distinguish the links despite $c_u^\ell = c_{u'}^\ell$ and $c_v^\ell = c_{v'}^\ell$, this implies that:

$$\left\{ \rho_1(w, G, \mathbf{X}) \,\middle|\, w \in \bigcup_{j=0}^{m} N^j(u, v) \right\} \neq \left\{ \rho_1(i, G, \mathbf{X}) \,\middle|\, i \in \bigcup_{j=0}^{m} N^j(u', v') \right\} \tag{1}$$

Since $k_\rho^{M_1} \leq k_\rho^{M_2}$, and both models operate on the same neighborhood radius $m$, it follows that there exists a number of layers (possibly greater than in $F_1$ [21, 25]) such that $F_2$ can also distinguish these sets. That is,

$$\left\{ \rho_2(w, G, \mathbf{X}) \,\middle|\, w \in \bigcup_{j=0}^{m} N^j(u, v) \right\} \neq \left\{ \rho_2(i, G, \mathbf{X}) \,\middle|\, i \in \bigcup_{j=0}^{m} N^j(u', v') \right\},$$

which implies

$$F_2((u, v), G, \mathbf{X}^0) \neq F_2((u', v'), G, \mathbf{X}^0).$$

Therefore, by Definition 2.8, $F_1 \leq F_2$.

□

***Proof of Theorem 3.3*. 1.** This follows directly from Theorem 3.2. Pure GNNs only rely on node representations $\phi$ and the combining function $g$, without any function $f$ over the $m$-order neighborhood. As established in Theorem 3.2, such models cannot distinguish between links whose endpoints are automorphic. In contrast, NCN, Neo-GNN, ELPH, and SEAL incorporate neighborhood-level functions $f$ (all with injective COMB operations, namely concatenation), which increases their expressive power.

**2.** According to Table 1, Neo-GNN uses a fixed number of message-passing layers $l^{\text{Neo-GNN}} = 1$, while NCN allows $l^{\text{NCN}}$ to be freely chosen. Since both models operate with the same type of base function $\rho$ (e.g., 1-WL), Theorem 3.2 implies that if $l^{\text{NCN}} \geq m^{\text{Neo-GNN}}$, then NCN can simulate or surpass Neo-GNN's receptive field and representational capability. Hence, Neo-GNN $\leq$ NCN.

**3.** By design, ELPH does not use any message-passing layers ($l^{\text{ELPH}} = 0$, see Table 1), and instead aggregates counts over the $m$-hop neighborhood. If Neo-GNN uses a neighborhood of size at least $m^{\text{ELPH}} - 1$, then Theorem 3.2 ensures it can access all the information available to ELPH and apply additional learned transformations via its GNN layers. Thus, Neo-GNN is more expressive.

**4.** ELPH relies only on aggregated neighborhood counts and uses $l^{\text{ELPH}} = 0$ layers. NCN uses a fixed radius $m^{\text{NCN}} = 1$ (Table 1), but allows a configurable number of layers. If $l^{\text{NCN}} \geq m^{\text{ELPH}} - 1$, then by Theorem 3.2 NCN can effectively capture structural signals that ELPH computes with wider but shallower architectures, achieving greater expressiveness.

**5.** SEAL is more expressive due to both its flexible architecture and enhanced message-passing capabilities. Specifically, SEAL builds a subgraph around the link and augments the adjacency matrix with positional encodings that are link-aware, effectively allowing a learned structural encoding over the link neighborhood $N^m(u, v)$. This corresponds to the expressive power of a higher-order WL test (namely 1-$|N^m(u, v)|$-WL [59]), which is strictly more powerful than 1-WL. Moreover, SEAL allows arbitrary $m^{\text{SEAL}}$ and $l^{\text{SEAL}}$, so for any other model with weaker $\rho$, one can always configure SEAL with the same $m$ and a sufficient number of layers to match or exceed its expressiveness. Prior results have shown that highly expressive architectures may require deeper message passing to converge [21, 25], which SEAL supports by design. □

# B   Additional Information on `LR-EXP`

## B.1   Evaluation method

This section provides a more detailed explanation of how the two components of the Reliable Paired Comparison work to evaluate different models in the synthetic setting.

**Major Procedure**   Given a graph $G$, we select two structurally distinct links $e = (u, v), e' = (u', v')$. Then we create $q$ isomorphic copies $\{G_1, G_2, \ldots, G_q\}$ by randomly permuting its nodes. For each copy $G_i$ and pair of links $e_i, e_i'$, we compute the embedding differences:

$$d_i = f(G_i, e_i) - f(G_i, e_i'), \quad i \in [q], \tag{15}$$

where $f$ is the embedding function learned by the model. Assuming that the difference vectors are independent random vectors $\mathcal{N}(\mu, \Sigma)$, we consider that $f(G_i)$ follows a Gaussian distribution, so random permutations only introduce Gaussian noise into the results.

If the model is not able to distinguish links $e$ and $e'$, the mean difference should be $\mu = 0$. To check this a $\alpha$−level Hotelling's T-square test is conducted, comparing hypothesis $H_0 : \mu = 0$ and $H_1 : \mu \neq 0$. The $T^2$−statistic for $\mu$ is calculated as:

$$T^2 = q(\bar{d} - \mu)^T S^{-1}(\bar{d} - \mu), \tag{16}$$

where

$$\bar{d} = \frac{1}{q}\sum_{i=1}^{q} d_i, \quad \mathbf{S} = \frac{1}{q-1}\sum_{i=1}^{q}(d_i - \bar{d})(d_i - \bar{d})^T. \tag{17}$$

Hotelling's T-square test establishes that $T^2$ follows the distribution of a random variable $\frac{(q-1)d}{q-d}F_{d,q-d}$, where $F_{d,q-d}$ represents an $F$-distribution with degrees of freedom $d$ and $q - d$ [23]. This theorem provides a link between the unknown parameter $\mu$ and a specific distribution $F_{d,q-d}$, enabling us to validate the confidence interval of $\mu$ by analyzing how well it matches the assumed distribution. To evaluate the hypothesis $H_0 : \boldsymbol{\mu} = 0$, we substitute $\boldsymbol{\mu} = 0$ into Equation (16), leading to:

$$T^2_{\text{test}} = q\bar{\boldsymbol{d}}^T S^{-1} \bar{\boldsymbol{d}}. \tag{18}$$

An $\alpha$-level test of $H_0 : \boldsymbol{\mu} = 0$ against the alternative $H_1 : \boldsymbol{\mu} \neq 0$ supports $H_0$ (meaning the model fails to differentiate the pair) if:

$$T^2_{\text{test}} = q\bar{\boldsymbol{d}}^T S^{-1} \bar{\boldsymbol{d}} < \frac{(q-1)d}{(q-d)} F_{d,q-d}(\alpha), \tag{19}$$

where $F_{d,q-d}(\alpha)$ represents the upper $(100\alpha)$-th percentile of the $F$-distribution $F_{d,q-d}$ [20], with degrees of freedom $d$ and $q - d$. Conversely, we reject $H_0$ (indicating that the model successfully distinguishes the pair) if:

$$T^2_{\text{test}} = q\bar{\boldsymbol{d}}^T S^{-1} \bar{\boldsymbol{d}} > \frac{(q-1)d}{(q-d)} F_{d,q-d}(\alpha). \tag{20}$$

**Reliability Check**   By selecting an appropriate value of $\alpha$, the Main Procedure ensures a reliable confidence interval to evaluate distinguishability. However, choosing $\alpha$ heuristically may not be the optimal approach. Additionally, computational precision can introduce deviations in the assumed Gaussian fluctuations. To address this issue, we employ the Reliability Check, which captures both external differences between two graphs and internal variations within a single graph.

Without loss of generality (WLOG), we replace $(G_i, e'_i)$ with a permutation of $G$, denoted as $(G^\pi, e^\pi)$. This allows us to analyze the internal fluctuations of $G$ for the same link. We follow the same steps as in the Main Procedure.

Proceeding with $(G_i, e_i)$ and $(G^\pi_i, e^\pi_i)$, we compute the $T^2$-statistic as:

$$T^2_{\text{reliability}} = q\bar{d}^T S^{-1} \bar{d}, \tag{21}$$

where

$$\bar{d} = \frac{1}{q} \sum_{i=1}^{q} d_i, \quad d_i = f(G_i, e_i) - f(G^\pi_i, e^\pi_i), \quad i \in [q], \tag{22}$$

$$S = \frac{1}{q-1} \sum_{i=1}^{q} (d_i - \bar{d})(d_i - \bar{d})^T. \tag{23}$$

Since $G$ and $G^\pi$ are isomorphic, the GNN should not differentiate between them, implying that $\mu = 0$. Consequently, the test is considered reliable only if

$$T^2_{\text{reliability}} < \frac{(q-1)d}{(q-d)} F_{d,q-d}(\alpha). \tag{24}$$

By combining the reliability check with the distinguishability results, we obtain the full RPC Reliability Procedure Check.

For each pair of structurally different links $e$ and $e'$, the threshold is computed as:

$$\text{Threshold} = \frac{(q-1)d}{(q-d)} F_{d,q-d}(\alpha). \tag{25}$$

Next, we perform the Main Procedure on $e$ and $e'$ to evaluate distinguishability, and we conduct the Reliability Check on $e$ and $e^\pi$. The GNN is considered capable of distinguishing $e$ and $e'$ only if both conditions

$$T^2_{\text{reliability}} < \text{Threshold} < T^2_{\text{test}} \tag{26}$$

are satisfied.

## B.2 Experimental Results with different significance level values

We tested the sensitivity of results of Section 4.3 to the choice of the significance level in the reliability check. Specifically, we experimented with three commonly used values in statistical testing, i.e., . The resulting performance on LR-EXP are reported in Table 4.

As expected, all standard GNN models consistently fail to recognize any links, regardless of the significance level. Among the more expressive models, the majority show stable behavior across different values of significance, with fluctuations that remain within the standard deviation. *BUDDY* and *NCN* exhibit slightly larger fluctuations in absolute value, but the key point for this experiment is the expressiveness ranking across models. This ranking remains robust across all significance levels, with one minor exception: at significance level 0.1, *Neo-GNN* slightly surpasses *NCN*. However, this difference falls within the standard deviation range.

Table 4: Test Precision (mean $\pm$ std over 5 runs) for significance level values $0.1, 0.05, 0.01$.

| Model | 0.1 | 0.05 | 0.01 |
|---|---|---|---|
| *GCN* | $0 \pm 0$ | $0 \pm 0$ | $0 \pm 0$ |
| *GAT* | $0 \pm 0$ | $0 \pm 0$ | $0 \pm 0$ |
| *GAE* | $0 \pm 0$ | $0 \pm 0$ | $0 \pm 0$ |
| *SAGE* | $11 \pm 0$ | $8 \pm 4$ | $3 \pm 0$ |
| *BUDDY* | $54 \pm 5$ | $45 \pm 1$ | $59 \pm 1$ |
| *ELPH* | $61 \pm 1$ | $62 \pm 7$ | $62 \pm 1$ |
| *Neo-GNN* | $76 \pm 6$ | $75 \pm 2$ | $71 \pm 2$ |
| *NCN* | $70 \pm 1$ | $75 \pm 1$ | $81 \pm 1$ |
| *SEAL* | $92 \pm 0$ | $97 \pm 0$ | $97 \pm 0$ |

## C Further Experiments on LR-EXP

In the experiment of Section 4.3, to ensure a fair comparison, we selected for each model the maximum neighborhood radius that makes it as expressive as possible while still respecting its design constraints. For example, in models like *NCN*, the radius is fixed by design and equal to 1, so it cannot be varied. In others, such as *ELPH*, the radius is tunable but is constrained to a maximum of 3 by the original authors for computational reasons, and we adopted that upper bound. As for GNN depth, we fixed the number of layers to 3 as commonly done in the literature [42].To explore this further, we conducted two additional experiments on the LR-EXP dataset: 1) **Fixed radius** $m = 1$, varying the number of layers (see Table 5); 2) **Fixed number of layers** $l = 3$, varying the neighborhood radius (see Table 6).

Table 5: 1) Fixed Radius = 1

| # Layers | *NCN* | *Neo-GNN* | *ELPH* | *SEAL* |
|---|---|---|---|---|
| 1 | $74 \pm 1$ | $42 \pm 4$ | $40 \pm 4$ | $79 \pm 3$ |
| 2 | $75 \pm 1$ | $42 \pm 4$ | $55 \pm 4$ | $89 \pm 1$ |
| 3 | $75 \pm 1$ | $45 \pm 3$ | $62 \pm 2$ | $89 \pm 2$ |

These results are fully consistent with Theorem 3.2 (Point 2): when fixing the radius, increasing the number of layers improves expressiveness in all the models; when fixing the number of layers, increasing the radius also improves expressiveness in all the models.

Table 6: 2) Fixed # Layers = 3

| Radius | *NCN* | *Neo-GNN* | *ELPH* | *SEAL* |
|---|---|---|---|---|
| 1 | $75 \pm 1$ | $45 \pm 3$ | $62 \pm 2$ | $89 \pm 2$ |
| 2 | — | $63 \pm 3$ | $61 \pm 4$ | $90 \pm 3$ |
| 3 | — | $75 \pm 2$ | $62 \pm 7$ | $97 \pm 0$ |

## D    Computational Costs

The overall time complexity of models for link prediction can be expressed as $O(B + Ct)$, where $t$ is the number of links to be predicted, $B$ is the preprocessing cost and $C$ per-link cost [46]. In Table 7 we report the time complexity of existing models, where $n$ is the number of nodes, $d$ is the maximum node degree, $h$ is the complexity of the hash function used in *BUDDY*, $f$ is the feature dimension and $m$ is the radius of the neighborhood.

Table 7: Time complexity of models expressed as $O(B + Ct)$, where $t$ is the number of links to be predicted, $B$ is the preprocessing cost and $C$ per-link cost.

| Method | B | C |
|---|---|---|
| GAE | $ndf + nf^2$ | $f^2$ |
| Neo-GNN | $ndf + nf^2 + nd^m$ | $d^m + f^2$ |
| BUDDY | $ndf + nh$ | $h + f^2$ |
| SEAL | $0$ | $d^{(m+1)}f + d^m f^2$ |
| NCN | $ndf + nf^2$ | $df + f^2$ |
| NCNC | $ndf + nf^2$ | $d^2 f + df^2$ |

Among these, *SEAL* is the most expressive but also the most computationally expensive, due to its exponential dependence on the neighborhood radius. *NCN* and *BUDDY* provide the best trade-off between expressiveness and computational cost. *Neo-GNN*, while more expensive than *NCN*, is in fact less expressive. *GAE* is the least costly, but also the least expressive.

## E    Experimental details

### E.1    Real world datasets

Table 8 presents the statistics of the standard datasets commonly used for link prediction tasks. The Cora, Citeseer, and Pubmed datasets are well-known citation networks, often employed as benchmarks for graph-based learning methods. These datasets are relatively small in scale, both in terms of nodes and edges. In contrast, the OGB (Open Graph Benchmark) datasets including ogbl-collab, ogbl-ddi, ogbl-ppa, and ogbl-citation2 are significantly larger and more complex, providing challenging benchmarks for evaluating scalability and performance on large graphs.

For Cora, Citeseer, and Pubmed, we adopt a fixed train/validation/test split of 85/5/10%. For the OGB datasets, we use the official splits provided by the OGB benchmark.

Table 8: Statistics of datasets. The split ratio is for train/validation/test.

| **Dataset** | Cora | Citeseer | Pubmed | ogbl-collab | ogbl-ddi | ogbl-ppa | ogbl-citation2 |
|---|---|---|---|---|---|---|---|
| #Nodes | 2,708 | 3,327 | 18,717 | 235,868 | 4,267 | 576,289 | 2,927,963 |
| #Edges | 5,278 | 4,676 | 44,327 | 1,285,465 | 1,334,889 | 30,326,273 | 30,561,187 |
| Mean Degree | 3.90 | 2.81 | 4.74 | 10.90 | 625.68 | 105.25 | 20.88 |
| Split Ratio | 85/5/10 | 85/5/10 | 85/5/10 | 92/4/4 | 80/10/10 | 70/20/10 | 98/1/1 |

Table 9 reports the statistics of additional datasets characterized by high symmetry, which are also used in our experiments. These datasets span various domains, including social networks, biological systems, and knowledge graphs, offering diverse structural properties and serving as valuable benchmarks for evaluating models under highly regular and repetitive connection patterns.

Specifically, the *AIFB*[5] dataset models the organizational structure of the AIFB research institute, capturing relationships among its staff, research groups, and publications. *Edit-TSW*[2] models user interactions and content editing activities within the Wiktionary platform. Finally, the *NSC*[6], *YST*[3], and *GRQ*[3] datasets, introduced in [12], focus on relational learning tasks and exhibit highly symmetric structures, making them particularly suitable for evaluating link prediction models.

Table 9: Statistics of datasets. The split ratio is for train/validation/test.

| Dataset | NSC | YST | GRQ | aifb | edit-tsw |
|---|---|---|---|---|---|
| #Nodes | 332 | 2,284 | 5,241 | 8,285 | 1,079 |
| #Edges | 2,126 | 6,646 | 14,484 | 46,042 | 2.756 |
| Mean Degree | 12.81 | 5.82 | 5.53 | 5.56 | 5.11 |
| Split Ratio | 80/10/10 | 80/10/10 | 80/10/10 | 80/10/10 | 80/10/10 |

## E.2 Implementation details

This section provides hyperparameters details for all models trained and evaluated on both `LR-EXP` and the real-world datasets. All experiments were conducted on a workstation running Ubuntu 22.04 with an AMD Ryzen 9 7950X CPU (32 threads), 124GB of RAM, and two NVIDIA GeForce RTX 4090 GPUs (24GB each).

**Synthetic Hyperparameter Settings**  We present the hyperparameter search space for the `LR-EXP` datasets in Table 10. For each model, we initially followed the hyperparameter configurations proposed in their respective papers. However, due to the small size of the synthetic graphs in `LR-EXP`, we significantly reduced the embedding dimensions, which led to improved performance in most cases.

Table 10: Hyperparameter Search Ranges for `LR-EXP`. Abbreviations: LR = Learning Rate, Drop. = Dropout, WD = Weight Decay, #L = Number of Model Layers, #P = Number of Prediction Layers, Dim = Embedding Dimension.

| Dataset | LR | Drop. | WD | #L | #P | Dim |
|---|---|---|---|---|---|---|
| `LR-EXP` | (0.01, 0.001) | (0.1, 0.3, 0.5) | (1e-4, 1e-7, 0) | (1, 2, 3) | (1, 2, 3) | (2–256) |

**Real-World Hyperparameter Settings**  We report the hyperparameter search space for all real-world datasets used in our experiments in Table 11. For *Cora*, *Citeseer*, *Pubmed*, *ogbl-collab*, *ogbl-ddi*, *ogbl-ppa*, and *ogbl-citation2*, we adopt the hyperparameter settings declared in the paper [32]. For the remaining datasets, *NSC*, *YST*, *GRQ*, *aifb*, and *edit-tsw*, we conducted a hyperparameter search using the ranges specified in the table.

## F  Real-world results with standard deviations

In this section, we present the same results as in Table 3, now including standard deviations computed over 5 runs with different random seeds. Specifically, Table 12 reports the MRR results on standard small datasets, Table 13 shows the MRR results on the OGB benchmark, and Table 14 reports the MRR results for highly symmetric datasets.

**Hits@k.**  We also report the Hits@$k$ metric, using $k = 10$ for the standard small datasets (Table 15) and for the new highly symmetric dataset (Table 16), and $k = 20$ for the larger datasets (Table 17), following the evaluation protocol of Li et al. [32].

---

[5]https://pytorch-geometric.readthedocs.io/
[6]https://github.com/LeiCaiwsu/LGLP/tree/main/LGLP/Python/data

Table 11: Hyperparameter search ranges for all real-world datasets. Abbreviations: LR = Learning Rate, Drop. = Dropout, WD = Weight Decay, #L = Number of Model Layers, #P = Number of Prediction Layers, Dim = Embedding Dimension.

| Dataset | LR | Drop. | WD | #L | #P | Dim |
|---|---|---|---|---|---|---|
| Cora | (0.01, 0.001) | (0.1, 0.3, 0.5) | (1e-4, 1e-7, 0) | (1, 2, 3) | (1, 2, 3) | (128, 256) |
| Citeseer | (0.01, 0.001) | (0.1, 0.3, 0.5) | (1e-4, 1e-7, 0) | (1, 2, 3) | (1, 2, 3) | (128, 256) |
| Pubmed | (0.01, 0.001) | (0.1, 0.3, 0.5) | (1e-4, 1e-7, 0) | (1, 2, 3) | (1, 2, 3) | (128, 256) |
| ogbl-collab | (0.01, 0.001) | (0, 0.3, 0.5) | 0 | 3 | 3 | 256 |
| ogbl-ddi | (0.01, 0.001) | (0, 0.3, 0.5) | 0 | 3 | 3 | 256 |
| ogbl-ppa | (0.01, 0.001) | (0, 0.3, 0.5) | 0 | 3 | 3 | 256 |
| ogbl-citation2 | (0.01, 0.001) | (0, 0.3, 0.5) | 0 | 3 | 3 | 128 |
| NSC | (0.01, 0.001) | (0.1, 0.3, 0.5) | (1e-4, 1e-7, 0) | (1, 2, 3) | (1, 2, 3) | (64, 128, 256) |
| YST | (0.01, 0.001) | (0.1, 0.3, 0.5) | (1e-4, 1e-7, 0) | (1, 2, 3) | (1, 2, 3) | (64, 128, 256) |
| GRQ | (0.01, 0.001) | (0.1, 0.3, 0.5) | (1e-4, 1e-7, 0) | (1, 2, 3) | (1, 2, 3) | (64, 128, 256) |
| aifb | (0.01, 0.001) | (0.1, 0.3, 0.5) | (1e-4, 1e-7, 0) | (1, 2, 3) | (1, 2, 3) | (64, 128, 256) |
| edit-tsw | (0.01, 0.001) | (0.1, 0.3, 0.5) | (1e-4, 1e-7, 0) | (1, 2, 3) | (1, 2, 3) | (64, 128, 256) |

Table 12: MRR with standard deviations on real-world link prediction datasets.

| Models | NSC | YST | GRQ | AIFB | EDIT-TSW |
|---|---|---|---|---|---|
| $\hat{r}_G$ | 0.16 | 0.24 | 0.30 | 0.41 | 0.67 |
| GCN | 26.79 ($\pm$ 1.08) | 1.17 ($\pm$ 0.02) | 6.89 ($\pm$ 0.47) | 10.85 ($\pm$ 0.12) | 4.12 ($\pm$ 0.21) |
| GAT | 2.47 ($\pm$ 3.58) | 0.40 ($\pm$ 0.00) | 0.63 ($\pm$ 0.40) | 1.07 ($\pm$ 0.51) | 0.40 ($\pm$ 0.00) |
| SAGE | 20.32 ($\pm$ 0.11) | 1.77 ($\pm$ 1.38) | 1.34 ($\pm$ 1.32) | 0.54 ($\pm$ 0.08) | 8.35 ($\pm$ 7.27) |
| GAE | 16.61 ($\pm$ 10.00) | 2.32 ($\pm$ 0.01) | 2.32 ($\pm$ 0.01) | 10.98 ($\pm$ 0.01) | 3.54 ($\pm$ 0.01) |
| BUDDY | 18.40 ($\pm$ 0.07) | 13.68 ($\pm$ 0.61) | 46.23 ($\pm$ 0.42) | 13.73 ($\pm$ 0.11) | 22.66 ($\pm$ 0.46) |
| ELPH | 26.09 ($\pm$ 1.04) | 13.15 ($\pm$ 1.12) | 38.88 ($\pm$ 0.55) | 13.39 ($\pm$ 0.60) | 10.10 ($\pm$ 7.37) |
| Neo-GNN | 22.86 ($\pm$ 0.75) | 5.78 ($\pm$ 0.12) | 29.51 ($\pm$ 0.38) | 13.11 ($\pm$ 0.51) | 8.10 ($\pm$ 0.29) |
| NCN | 30.36 ($\pm$ 0.28) | 11.99 ($\pm$ 0.06) | 48.45 ($\pm$ 0.02) | 6.47 ($\pm$ 0.23) | 10.32 ($\pm$ 0.39) |
| NCNC | 30.20 ($\pm$ 0.03) | 13.54 ($\pm$ 0.47) | 47.60 ($\pm$ 0.52) | OOM | 10.30 ($\pm$ 0.39) |
| SEAL | 30.85 ($\pm$ 1.46) | 17.51 ($\pm$ 0.94) | 56.72 ($\pm$ 1.35) | 16.30 ($\pm$ 0.22) | 25.82 ($\pm$ 1.47) |

Table 13: MRR with standard deviations on real-world link prediction datasets.

| Models | CORA | CITESEER | PUBMED |
|---|---|---|---|
| $\hat{r}_G$ | 0.02 | 0.02 | 0.08 |
| GCN | 16.61 ($\pm$ 0.30) | 21.09 ($\pm$ 0.88) | 7.13 ($\pm$ 0.27) |
| GAT | 13.84 ($\pm$ 0.68) | 19.58 ($\pm$ 0.84) | 4.95 ($\pm$ 0.14) |
| SAGE | 14.74 ($\pm$ 0.69) | 21.09 ($\pm$ 1.15) | 9.40 ($\pm$ 0.70) |
| GAE | 18.32 ($\pm$ 0.41) | 25.25 ($\pm$ 0.82) | 5.27 ($\pm$ 0.25) |
| BUDDY | 13.71 ($\pm$ 0.59) | 22.84 ($\pm$ 0.36) | 7.56 ($\pm$ 0.18) |
| Neo-GNN | 13.95 ($\pm$ 0.39) | 17.34 ($\pm$ 0.84) | 7.74 ($\pm$ 0.30) |
| NCN | 14.66 ($\pm$ 0.95) | 28.65 ($\pm$ 1.21) | 5.84 ($\pm$ 0.22) |
| NCNC | 14.98 ($\pm$ 1.00) | 24.10 ($\pm$ 0.65) | 8.58 ($\pm$ 0.59) |
| SEAL | 10.67 ($\pm$ 3.46) | 13.16 ($\pm$ 1.66) | 5.88 ($\pm$ 0.53) |

Table 14: MRR with standard deviations on real-world link prediction datasets.

| Models | OGBL-CITATION2 | OGBL-DDI | OGBL-PPA | OGBL-COLLAB |
|---|---|---|---|---|
| $\hat{r}_G$ | 0.01 | 0.10 | 0.12 | 0.13 |
| GCN | 19.98 ($\pm$ 0.35) | 13.46 ($\pm$ 0.34) | 26.94 ($\pm$ 0.48) | 6.09 ($\pm$ 0.38) |
| GAT | 22.05 ($\pm$ 0.12) | 12.92 ($\pm$ 0.39) | 27.27 ($\pm$ 0.30) | 4.18 ($\pm$ 0.33) |
| SAGE | OOM | 12.60 ($\pm$ 0.72) | OOM | 5.53 ($\pm$ 0.50) |
| GAE | OOM | 3.49 ($\pm$ 1.73) | OOM | OOM |
| BUDDY | 19.17 ($\pm$ 0.20) | 12.43 ($\pm$ 0.50) | 27.00 ($\pm$ 0.33) | 5.67 ($\pm$ 0.36) |
| Neo-GNN | 16.12 ($\pm$ 0.25) | 10.86 ($\pm$ 2.16) | 21.68 ($\pm$ 1.14) | 5.23 ($\pm$ 0.90) |
| NCN | 23.35 ($\pm$ 0.28) | 12.86 ($\pm$ 0.78) | 35.06 ($\pm$ 0.26) | 5.09 ($\pm$ 0.38) |
| NCNC | 19.61 ($\pm$ 0.54) | >24h | 33.52 ($\pm$ 0.26) | 4.73 ($\pm$ 0.86) |
| SEAL | 20.60 ($\pm$ 1.28) | 9.99 ($\pm$ 0.90) | 29.71 ($\pm$ 0.71) | 6.43 ($\pm$ 0.32) |

Table 15: Hits@10 with standard deviations on real-world link prediction datasets.

| Models | NSC | YST | GRQ | AIFB | EDIT-TSW |
|---|---|---|---|---|---|
| $\hat{r}_G$ | 0.16 | 0.24 | 0.30 | 0.41 | 0.67 |
| GCN | 43.08 (± 1.36) | 0.75 (± 0.00) | 13.47 (± 2.97) | 17.56 (± 0.92) | 5.91 (± 0.97) |
| GAT | 2.36 (± 4.09) | 0.00 (± 0.00) | 0.44 (± 0.76) | 1.51 (± 1.73) | 0.00 (± 0.00) |
| SAGE | 24.06 (± 3.68) | 3.56 (± 3.96) | 2.44 (± 4.17) | 0.41 (± 0.38) | 10.67 (± 10.07) |
| GAE | 30.38 (± 9.90) | 3.38 (± 0.00) | 3.38 (± 0.00) | 16.92 (± 0.09) | 3.80 (± 0.00) |
| BUDDY | 28.93 (± 2.68) | **22.89** (± 0.69) | 67.26 (± 0.25) | 24.28 (± 0.57) | 36.00 (± 2.27) |
| ELPH | 38.37 (± 0.72) | 22.34 (± 1.15) | 59.44 (± 0.99) | **26.19** (± 1.83) | 21.45 (± 15.19) |
| Neo-GNN | 37.11 (± 0.72) | 10.29 (± 0.17) | 44.50 (± 0.24) | **24.29** (± 1.74) | 12.36 (± 1.09) |
| NCN | **47.17** (± 0.94) | 20.03 (± 0.00) | **68.67** (± 0.14) | 9.42 (± 0.76) | 16.48 (± 0.56) |
| NCNC | **46.86** (± 0.27) | **22.44** (± 1.04) | **68.28** (± 0.23) | OOM | 15.03 (± 1.87) |
| SEAL | **51.26** (± 0.98) | **28.36** (± 1.46) | **71.64** (± 1.04) | **31.84** (± 0.70) | **42.55** (± 2.39) |

Table 16: Hits@10 with standard deviations on real-world link prediction datasets.

| Models | CORA | CITESEER | PUBMED |
|---|---|---|---|
| $\hat{r}_G$ | 0.02 | 0.02 | 0.08 |
| GCN | **36.26** (± 1.14) | 47.23 (± 1.88) | 15.22 (± 0.57) |
| GAT | 32.89 (± 1.27) | 45.30 (± 1.30) | 9.99 (± 0.64) |
| SAGE | 34.65 (± 1.47) | 48.75 (± 1.85) | **20.54** (± 1.40) |
| GAE | **37.95** (± 1.24) | **49.65** (± 1.48) | 10.50 (± 0.46) |
| BUDDY | 30.40 (± 1.18) | **48.35** (± 1.18) | 16.78 (± 0.53) |
| Neo-GNN | 31.27 (± 0.72) | 41.74 (± 1.18) | 17.88 (± 0.71) |
| NCN | 35.14 (± 1.04) | **53.41** (± 1.46) | 13.22 (± 0.56) |
| NCNC | **36.70** (± 1.57) | 53.72 (± 0.97) | **18.81** (± 1.16) |
| SEAL | 24.27 (± 6.74) | 27.37 (± 3.20) | 12.47 (± 1.23) |

Table 17: Hits@20 with standard deviations on real-world link prediction datasets.

| Models | OGBL-CITATION2 | OGBL-DDI | OGBL-PPA | OGBL-COLLAB |
|---|---|---|---|---|
| $\hat{r}_G$ | 0.01 | 0.10 | 0.12 | 0.13 |
| GCN | **51.72** (± 0.46) | 64.76 (± 1.45) | 68.38 (± 0.73) | 22.48 (± 0.81) |
| GAT | **53.13** (± 0.15) | **66.83** (± 2.23) | 69.49 (± 0.43) | 18.30 (± 1.42) |
| SAGE | OOM | **67.19** (± 1.18) | OOM | 21.26 (± 1.32) |
| GAE | OOM | 17.81 (± 9.80) | OOM | OOM |
| BUDDY | 47.81 (± 0.37) | 58.71 (± 1.63) | 71.50 (± 0.68) | **23.35** (± 0.73) |
| Neo-GNN | 43.17 (± 0.53) | 51.94 (± 10.33) | 64.81 (± 2.26) | 21.03 (± 3.39) |
| NCN | **53.76** (± 0.20) | **65.82** (± 2.66) | **81.89** (± 0.31) | 20.84 (± 1.31) |
| NCNC | 51.69 (± 1.48) | >24h | **82.24** (± 0.40) | 20.49 (± 3.97) |
| SEAL | 48.62 (± 1.93) | 49.74 (± 2.39) | **76.77** (± 0.94) | 21.57 (± 0.38) |

