# OpenReview forum: "Bridging Theory and Practice in Link Representation with Graph Neural Networks"
_NeurIPS.cc/2025/Conference — NeurIPS 2025 spotlight_

### Official Review · Reviewer_i6RA · 2025-06-06

**Clarity:** 3
**Significance:** 3
**Originality:** 4
**Rating:** 5
**Confidence:** 4

**Summary:**

This paper studies the expressiveness hierarchy of the link prediction models. Beyond previous k-WL test that focuses on node-level expressiveness, this work establishes a hierarchy that can describes the level of expressiveness for recent SOTA link prediction models. By introducing a framwork, this paper unifies common link prediction methods so that their expressiveness can be compared directly under this framework. Then the study introduces a synthetic benchmark, specifically designed to test the link-level expressiveness for link prediction models. Several interesting observations have been made on both the synthetic benchmark and the real-world datasets.

**Questions:**

1. When defining the neighborhood of a node $N^m(v)$, the paper defines as $m$-far away from $v$. How is the distance metric defined here? Is shortest-path or "number-of-steps" distance employed here?

2. When computing the symmetric metrics, is the node feature considered or only the graph structure? If the node feature is considered, how is the WL-test approximation applied to assign the inital colors?

**Ethical Concerns:**

["NO or VERY MINOR ethics concerns only"]

**Final Justification:**

My questions and concerns were fully addressed. Therefore, I update my score towards accept.

**Limitations:**

See weakness.

**Quality:**

4

**Strengths And Weaknesses:**

Strengths:

[S1] The expressiveness hierarchy of link-level representation by GNNs is an open question in the field. The paper provides a unfied framework to include common link predictors and can depict their expressiveness under this framework. This study provides a good insight of how/why the GNN4LP, especially those with SF, performs well on current benchmark.

[S2] The observations made in experimental sections are interesting. The empirical results on synthetic graphs well align with the theorectical conclusions. Besides, the results on real-world graphs shows the distinct strengths of different link predictors.

Weakness:

[W1] The proposed framework only works for GNNs, especially the message-passing types. There are many other link prediction methods without message-passing but performs well. For instance, graph transformer, LGLP[1], network embedding methods and typical heuristic methods like Common Neighbor families, Katz Index, Page Rank Index are not discussed in the paper.

[W2] The notation should be consistent. For example in Definition 3.1, $h(u, v, G, X^0)$ does not include the graph structure $G$ in equation (7).

[1]  Cai, Lei, et al. "Line graph neural networks for link prediction."

---

> ### Author Rebuttal · Authors · 2025-07-30
>
> **We would like to thank the reviewer for the thoughtful comments and useful suggestions.**
>
> ---
>
> **W1: The proposed framework only works for GNNs**
>
> We agree that many effective link prediction methods go beyond message-passing, including transformer-based models and heuristic methods. However, our focus is intentional: understanding the expressiveness of MP-based link representations is a necessary first step, just as MP models have been the starting point for theoretical analysis in graph-level tasks. These foundational studies have later inspired and supported the development of more powerful models such as graph transformers.
>
> We believe that the same applies to link representation: MP models are still not fully understood in this context, and our framework provides the first steps toward filling that gap. As we state in the paper, in the Limitations section (Section 6), extending the framework to encompass transformer-based architectures is our intention for future work.
>
> ---
>
> **W2: The notation should be consistent. For example in Definition 3.1 h does not include the graph structure in equation (7).**
>
> For ease of readability, we have omitted the graph $G $ as an explicit argument in message-passing and link-representation functions, assuming it is implicitly defined as the underlying structure on which these functions operate. We have specified it in line 159-161 of the paper. To maintain the consistency, we will follow this notation throughout the paper starting from line 159.
>
> ---
>
>
> **Q1: When defining the neighborhood of a node $N^m(v) $  the paper defines as m-far away from $v $. How is the distance metric defined here?**
>
> We thank the reviewer for the question. We would like to clarify that in Definition 2.2 we do not define a distance metric between nodes. Specifically, we do not assign a single distance value (e.g., shortest-path distance) between two nodes.
>
> Instead, we define the $ \mathcal{N}^m(v) $ neighborhood of a node $ v $ as the set of all nodes that are connected to $ v $ by a path of length $m $. Therefore, a node $ u $ may appear in multiple neighborhoods $ \mathcal{N}^m(v) $ for different values of $m $, whenever there exist multiple paths of different lengths $m$ connecting $u $ and $ v $.
>
> We will clarify this point in the paper by expanding the explanation following Definition 2.2.
>
> ---
>
> **Q2: When computing the symmetric metrics, is the node feature considered or only the graph structure? If the node feature is considered, how is the WL-test approximation applied to assign the initial colors?**
>
> Yes, node features are considered when computing the symmetry metric. We initialize the node colors based on the input features using an injective mapping from the feature space to the color space. In particular, we define each node’s initial color as:
>
> $c_v^0 = {HASH}(x_v) $
>
> where $x_v \in \mathbb{R}^f $ is the feature vector of node $v $, and `HASH` is an injective function mapping the features to unique colors.
>
> This initialization corresponds to iteration 0 of the WL test. From that point on, standard WL refinement proceeds, updating each node’s color based on its own color and the multiset of its neighbors’ colors. We will make this formalism explicit in the final version of the paper.

---

> > ### Comment · Reviewer_i6RA · 2025-08-01
> >
> > Thank the authors for addressing my questions. I still have questions about how the HASH function maps a multi-dimension with float values (continuous) feature vector to a discrete color. Can you explain more in details about it?

---

> > > ### Author Response · Authors · 2025-08-02
> > >
> > > We thank the reviewer for the follow-up and for engaging in this discussion.
> > >
> > > We note that, in practice, node features are stored in floating point format and therefore have finite precision. To map a multi-dimensional feature vector with float values to a discrete color, we proceed as follows. Let $\mathbf{x}_v$ be the feature vector of a node $v$. We construct its initial color $c_v^0$ by converting each feature of the vector to a string (using a sufficient number of decimal digits to preserve its precision), and then concatenating these string representations using a delimiter (a comma) between each feature (ex. $\mathbf{x}_v=[1.1, 0.2, 3.4]$ is converted to the string $c_v^0=$ " $1.1,0.2,3.4$ "). This results in a unique string representation of the feature vector, which is then used as the initial color in the WL algorithm. Standard WL hashing is then applied on top of these initial colors represented as strings. We will clarify this point explicitly in the revised version of the paper.
> > >
> > > As stated in Definition 2.1, we work with finite graphs with $n$ nodes, so there is only a finite number of feature vectors in any given graph. Even when feature vectors are theoretically considered continuous, as long as they are finite in number, they can be injectively mapped to discrete colors.

---

> > > > ### Comment · Reviewer_i6RA · 2025-08-03
> > > >
> > > > Thank the authors for detailing the coloring process. I have further questions about it.
> > > >
> > > > Since every node has different feature vector, will it make every node different initial color? if so,  there’s no symmetric at all in the graph because every node has its own orbit.

---

> > > > > ### Author Response · Authors · 2025-08-03
> > > > >
> > > > > We thank the reviewer for the question. The node features we consider are not node identifiers, so it is possible for different nodes to have the same feature vector. For example, in the OGB-PPA dataset, node features are 58-dimensional binary vectors where only one position is set to 1, indicating the species of the corresponding protein. As a result, all proteins from the same species share the same feature vector. In Cora, instead, node features are based on a bag-of-words representation. In this case, repeated feature vectors do exist but are rare — and indeed, Cora exhibits very low symmetry.
> > > > >
> > > > > Importantly, we note that the datasets with the highest symmetry — namely NSC, YST, GRQ, AIFB, and EDIT-TSW — have no initial node features, meaning that WL is initialized using node degrees. We will clarify this point in the revised version of the paper, explicitly listing which datasets include node features and explaining how this impacts the value of the symmetry measure.

---

> > > > > ### Author Response · Authors · 2025-08-06
> > > > >
> > > > > We sincerely thank the reviewer for the helpful discussion. We hope that our clarifications have addressed your concerns effectively.
> > > > >
> > > > > If there are any remaining doubts, suggestions for improvement, or points that you feel would benefit from further clarification, we would be very happy to address them. We would greatly appreciate any additional comments you might have including anything that could positively impact your score.
> > > > >
> > > > > Thank you again for your time and thoughtful engagement.

---

### Official Review · Reviewer_mA2W · 2025-06-30

**Clarity:** 3
**Significance:** 3
**Originality:** 3
**Rating:** 5
**Confidence:** 4

**Summary:**

This study presents a systematic study of the expressive power of MPNNs (message-passing type of GNNs) when they are used to generate link-level features. The authors proposes a formal framework that characterises any link representation model by the WL strength of the endpoint encoder $\phi$, the neighborhood encoder $\rho$ and the radius of neighborhood. Under this framework, they prove a partial order over several SOTA link predictors and obtain theorems. Beyond the theorectical foundation, they propsoe LR-EXP, a synthetic benchmark containing pairs of non-automorphic links whose endpoints are automorphic, and they introduce a symmetry score that approximates the density of node orbits in a real graph. Experiments confirm the predicted hierarchy on both LR-EXP and several real benchmarks, showing that SEAL is in practice and in theory the highly expressive models.

**Questions:**

1. In Section 3 the assumption of Thm 3.2 relies on COMB being injective under certain condition. Many practical implementations concatenate and then feed an MLP; is this always injective in training (for instance Neo-GNN), and if not, how sensitive are the conclusions?

**Ethical Concerns:**

["NO or VERY MINOR ethics concerns only"]

**Final Justification:**

I maintain my original score of Accept.

**Limitations:**

See the question above, and It would also be helpful for authors to discuss societal impact (if any) of their work.

**Quality:**

4

**Strengths And Weaknesses:**

Strengths

1. The study looks into an important and open problem—how to compare link-level expressiveness—using rigorous definitions and proofs. The formalism in the study is compact yet flexible enough to encode every mainstream message-passing link predictor, which makes the hierarchy both transparent and extensible.

2. Theorem 3.2 clearly exposes why using only endpoint embeddings is fundamentally limited, and why trading neighbourhood size for depth or higher-order WL power can be more efficient.

3. The synthetic benchmark LR-EXP is well-motivated. By forcing the endpoints of each test pair to share WL colours, the benchmark show that the empirical relationship between the symmetry score and test performance and it is convincing and offers a useful rule of thumb for practitioners when building GNN4LP.

Weakness

1. The framework is only restricted to message-passing designs. Graph transformer-based link predictors, line-graph methods such as LGLP and classical proximity indices like Katz Index are only discussed qualitatively; their formal place in the hierarchy therefore remains unclear.
2. In Section 4 the LR-EXP graphs contain at most seventeen nodes (Erdos–Rényi graph); while size is not essential to the expressiveness argument, I wonders whether the training protocol and Reliable Paired Comparison evaluation would remain tractable on larger graphs with rich attributes.
3. The real-world study focuses on ranking metrics like Hits@50, MRR. But the computational cost of the most expressive models (especially SEAL’s sub-graph extraction) is not reported; this is crucial for users balancing accuracy against throughput.

---

> ### Author Rebuttal · Authors · 2025-07-30
>
> **We would like to thank the reviewer for the thoughtful comments and useful suggestions.**
>
> ---
>
>
> **W1: The framework is only restricted to message-passing designs.**
>
> We agree that many effective link prediction methods go beyond message-passing, including transformer-based models and heuristic methods. However, our focus is intentional: understanding the expressiveness of MP-based link representations is a necessary first step, just as MP models have been the starting point for theoretical analysis in graph-level tasks. These foundational studies have later inspired and supported the development of more powerful models such as graph transformers.
> We believe that the same applies to link representation: MP models are still not fully understood in this context, and our framework provides the first steps toward filling that gap. As we state in the paper, in the Limitations section (Section 6), extending the framework to encompass transformer-based architectures is our intention for future work.
>
> ---
>
> **W2: Do training protocol and Reliable Paired Comparison evaluation would remain tractable on larger graphs with rich attributes?**
>
> Our training protocol uses a Siamese architecture where both branches share the same weights and apply the link representation model under evaluation. As a result, the computational cost of training is essentially the same as training the base model itself. Our framework does not introduce any additional significant overhead; the scalability depends entirely on the model being tested.
>
> The same applies to the Reliable Paired Comparison evaluation. After training, inference is performed on isomorphic copies of the original graph. These copies are generated with linear complexity in the graph size ($\mathcal{O}(|V| + |E|) $), and the inference time again depends only on the model used.
>
> In short, the scalability of our approach on larger graphs depends solely on the scalability of the link representation models being evaluated, not on the protocol itself.
>
> ---
>
> **W3: Missing computational cost of the most expressive models**
>
> We agree that computational cost is an important factor when evaluating expressive models. Below, we report the time complexity of the methods considered in our paper, as presented in [1]. The overall time complexity for link prediction is $\mathcal{O}(B + Ct) $, where t is the number of links to be predicted and the terms $B $ (preprocessing) and $C $ (per-link cost) are summarized below:
>
> | Method   | B                                 | C                               |
> |----------|-----------------------------------|---------------------------------|
> | GAE      | $ndf + nf^2$                       | $f^2$                            |
> | Neo-GNN  | $ndf + nf^2 + nd^{\ell}$               | $d^{\ell}+ f^2$                      |
> | BUDDY    | $ndf + nh$                       | $h + f^2$                       |
> | SEAL     | $0$                               | $d^{(\ell+1)}f + d^{\ell}f^2$           |
> | NCN      | $ndf + nf^2$                       | $df + f^2$                       |
> | NCNC     | $ndf + nf^2$                       | $d^2f + df^2$                     |
>
> with:
> - $n $: number of nodes in the graph
> - $d $: maximum node degree
> - $f $: feature dimension
> - $h$: complexity of the hash function used in BUDDY
> - $\ell $: neighborhood radius
>
> Among these, **SEAL** is the most expressive but also the most computationally expensive, due to its exponential dependence on the neighborhood radius $\ell$. **NCN** and **BUDDY** provide the best trade-off between expressiveness and computational cost. **Neo-GNN**, while more expensive than NCN, is in fact less expressive. **GAE** is the least costly, but also the least expressive.
>
> We will include this analysis in the final version of the paper.
>
> [1] Wang, Xiyuan, Haotong Yang, and Muhan Zhang. "Neural Common Neighbor with Completion for Link Prediction." The Twelfth International Conference on Learning Representations.
>
> ---
>
> **Q1: the assumption of Thm 3.2 relies on COMB being injective under certain condition. Many practical implementations concatenate and then feed an MLP; is this always injective in training (for instance Neo-GNN), and if not, how sensitive are the conclusions**
>
> We would like to clarify that the `COMB` function referenced in Theorem 3.2 specifically corresponds to the one defined in our Definition 3.1, that is, the function that combines the link endpoints’ representations with the representation of the neighborhood around the two nodes involved in the link. This should not be confused with the more general "combine" operations used inside GNN layers.
>
> As shown in Table 1 of the paper, all existing SOTA models (including Neo-GNN) implement the `COMB` function of Definition 3.1 as a simple concatenation, without any MLP. Since concatenation is inherently injective, the assumption in Theorem 3.2 holds for these models.
>
> If one implements `COMB` using both concatenation and an MLP, injectivity is no longer guaranteed. Since injectivity of `COMB` is an assumption of Theorem 3.2, if that assumption does not hold, the expressiveness guarantees derived from the theorem may no longer apply.

---

> > ### Comment · Reviewer_mA2W · 2025-08-01
> >
> > Thanks. I am satisfied by the responses and will keep my score.

---

### Official Review · Reviewer_qEnu · 2025-07-01

**Clarity:** 4
**Significance:** 3
**Originality:** 3
**Rating:** 5
**Confidence:** 5

**Summary:**

Graph neural networks have been enjoyed significant use in modeling graph structured data for tasks such as graph classification, node classification, and link prediction. Over the years, there are many different GNN architectures that have been introduced for these three diverse tasks; and the community has come to understand the representational capacity, or expressive power, of these models by characterizing the shape of data that they can learn to represent. In the case of graph and node classification, the traditional route to do this is through an appeal to the Weisfeiler Lehman graph isomorphism test. This has led the community to develop ever-more expressive models that, and ultimately, the "solving" of many significant graph benchmarks in the field. Link prediction has been comparatively less studied -- the automorphic node problem has been identified, and addressing it directly has led to techniques such as labeling tricks or models such as BUDDY/ELPH, and expressivity has been explored on a case-by-case basis. What has been missing in the literature is a hierarchy of link-level representations, which would allow the field to contextualize their models more completely, but to also begin to characterize the design principles going forward. The authors seek to address this task directly. They do so by introducing a unifying expressiveness framework and place existing models within it. They validate this new definition of expressivity in synthetic graphs generated with high degrees of symmetry, and verify the expressiveness gains. Ultimately, however, they observe that despite the significant improvements in expressivity, the gains on benchmark datasets are rather modest. The authors work ultimately raises more questions than it answers, but in doing so arms the field a new set of tools to answer these questions.

**Questions:**

1. The static link prediction problem is often a relatively contrived version of the industrial problem, because in industrial settings the graph is dynamic. Have you given any thought to how this work might extend to the dynamic link representation problem?
2. While many graphs are undirected (eg, the FB Friendship graph); many graphs _are_ directed (eg, a financial transaction graph). How would you extend this work to directed settings? Or are directed real-world graphs usually so lacking in symmetry that the link-level representations are easier to distinguish?
3. Could you share more about the intuition behind the loss? I suppose I would have expected to see a margin term
4. Have you tried to make a model like SAGE more expressive in your framework? You could do this for example by adding in DRNL features, positional encodings, and/or structural features (eg common neighbor counts). It would be highly interesting to me to see the performance of a simple model like SAGE as a function of expressivity improvements.
5. In Srinivasan et al [1], they show in remark 2 that graphsage provides node embeddings rather than structural representations due to the random walks. It seems to me like that would lead to SAGE potentially outperforming GCN on LR-EXP. How did you train SAGE? Did you sample the neighborhoods? How do you think about squaring this with your expressiveness hierarchy?
6. How well does the approximation in eq 12 agree with 11?
7. How sensitive are the results to the choice of α in the reliability check? Characterizing this would be helpful.

[1] https://arxiv.org/pdf/1910.00452

**Ethical Concerns:**

["NO or VERY MINOR ethics concerns only"]

**Final Justification:**

The work is technically valuable and provides important insights into the field of link prediction. I feel as if the paper is a clear accept, and this is only more true given the answers provided in the rebuttal.

**Limitations:**

yes

**Quality:**

4

**Strengths And Weaknesses:**

**Strengths**
1. The work is well motivated and addresses a significant problem in the space.
2. The work is well written and presented. There are no significant typos, and is relatively easy to follow despite the mathematically intricate nature of the work.
3. LR-EXP is a well constructed, albeit small, synthetic playground that provides meaningful information about performance of these models.
4. Theorem 3.2 is succinct and yet intuitive.
5. The synthetic experiments clearly display the ordering yielded by theorem 3.2, indicating that the theoretical framework is valid.
6. The experiments presented in Table 3 tell a relatively clear story that further justifies the theoretical framework.

**Weaknesses**
1. Minor typo -- in Table 3 some of the entries use a comma instead of a period.
2. The authors do not make much contact with previous link-representation expressivity results such as Srinivasan et al.
3. The authors do not spend much time explaining _why_ the symmetry definition given in eq 11 (or 12) would meaningfully impact the real-world expressivity of the models. I know that this isn't the usual way to consider this task, but it would be interesting to understand the required expressivity for a given task/dataset in terms of the available automorphisms in the graph at hand.
4. The gap between synthetic performance differences and real-world gains could be explored more deeply

---

> ### Author Rebuttal · Authors · 2025-07-30
>
> **We would like to thank the reviewer for the thoughtful comments and useful suggestions.**
>
> ---
>
> **W1: Minor typo -- in Table 3 some of the entries use a comma instead of a period.**
>
> We apologize for the formatting error, we will correct the table to use consistent notation.
>
> ---
>
> **W2: The authors do not make much contact with Srinivasan et al.**
>
> We fully agree that Srinivasan et al. provide a foundational result: they show that structural node representations produced by standard GNNs are not sufficient for link prediction, as link-level tasks require joint structural representations. This insight has led to a growing body of work on message-passing models that aim to overcome this limitation, often by incorporating joint structural features into the link representation.
> However, despite this growing literature, there is currently no unified framework for systematically analyzing and comparing the expressiveness of these models. This is precisely the gap our work aims to fill.
> Our main contribution is to introduce a general framework that subsumes all MP-based link representation models — from standard GNNs to more expressive models that use joint structural information. The framework allows these methods to be described using a common formal scheme, making it possible to compare their expressiveness in a principled way. We will clarify this connection to Srinivasan et al. more explicitly in the revised version of the paper.
>
> ---
>
> **W3: Why the symmetry definition given in eq 11 (or 12) would meaningfully impact the real-world expressivity of the models.**
>
> The symmetry metric directly reflects the amount of available automorphisms in the graph. When the symmetry is low, the graph structure is irregular, and standard GNNs are unlikely to fail, since structurally different links are more easily distinguishable. In contrast, high symmetry increases the chance of presence of structurally different links with identical representations given by standard GNNs.
> This is exactly what we observe in Table 3: on real-world datasets with low symmetry, standard GNNs perform well; on highly symmetric datasets, they fail to rank among the top-performing methods. We will clarify this connection more explicitly in the paper.
>
> ---
>
> **W4: The gap between synthetic performance differences and real-world gains could be explored more deeply**
>
> The synthetic dataset was intentionally designed to stress-test the expressiveness of the models, by including links that are particularly challenging to be distinguished by standard GNNs. This design amplifies the gap between expressive models (those using joint structural features) and standard GNNs.
>
> In real-world datasets, such extreme cases are less common, and high-symmetry patterns are harder to find. This is precisely what we show in our real-data experiments: expressive models still tend to perform better, but the gap is smaller – reflecting the lower frequency of hard-to-distinguish links in practice.
>
> Overall, our results confirm that expressiveness matters most when symmetry is present, and that the synthetic and real-world results are aligned in that regard.
>
> ---
>
> **Q1: Have you given any thought to how this work might extend to the dynamic link representation problem?**
>
> Our framework can be extended to the temporal link prediction setting by including the time variable $t $ as an explicit argument of the functions $\phi, \rho, h$ within the framework. This would allow the framework to cover temporal GNNs operating on both snapshot-based and event-based dynamic graphs [1].
>
> A particularly interesting direction would be to extend the analysis in [2] to the temporal link prediction setting and use our framework to investigate the expressiveness of time-then-graph and time-and-graph architectures in the temporal link prediction task. We see this as a promising line of future work.
>
> [1] Longa, Antonio, et al. "Graph Neural Networks for Temporal Graphs: State of the Art, Open Challenges, and Opportunities." Transactions on Machine Learning Research.
> [2] Gao, Jianfei, and Bruno Ribeiro. "On the equivalence between temporal and static equivariant graph representations." International Conference on Machine Learning. PMLR, 2022.
>
> ---
>
> **Q2: How would you extend this work to directed settings?**
>
> Our framework could be quite easily generalized to directed graphs by redefining the functions $h $ and $\psi $ to operate over ordered node pairs $(u, v) $, rather than treating the two nodes separately.
>
> ---
>
> **Q3: Could you share more about the intuition behind the loss? I suppose I would have expected to see a margin term**
>
> We agree that contrastive loss functions are often defined with an explicit margin term $\gamma $, which controls the degree of dissimilarity enforced between negative pairs. In our case, the goal is to fully minimize similarity between representations of non-automorphic links. Following the approach in [3], we deliberately adopted a zero-margin formulation to push the model toward maximal dissimilarity between such pairs. This choice aligns with our objective of measuring and testing expressiveness, where even small overlaps in representation between structurally different links would be undesirable.
>
> We will clarify this design choice in the revised paper.
>
> [3] Wang, Yanbo, and Muhan Zhang. "An Empirical Study of Realized GNN Expressiveness." Forty-first International Conference on Machine Learning.
>
> ---
>
> **Q4-Q5: Expressiveness of SAGE**
>
> We thank the reviewer for pointing this out. In our original implementation of GraphSAGE, we did not include neighborhood sampling, rather we used a deterministic aggregation over all neighbors. Following your observation and Remark 2 in Srinivasan et al., we re-ran the experiments using stochastic neighbor sampling. As expected, GraphSAGE now achieves an accuracy of 8.2 ± 3.5, outperforming GCN. We will include this updated result in the paper, along with an explicit reference to the theoretical motivation included in Srinivasan et al.
>
> ---
>
> **Q6: How well does the approximation in eq 12 agree with 11?**
> The approximation in Equation 12 reflects how closely the WL-induced partition (used in Equation 11) matches the true orbit partition. Although there is no complete characterization of WL failures in distinguishing nodes from different orbits, it is said that WL gives different representations to **almost all** nodes that belong to different orbits [4].
>
> A practical example is presented in [5], where the authors explicitly compare WL partitions and orbit partitions on the Alchemy dataset [6]. Out of 202,579 graphs, only in 4 graphs the partition of the nodes induced by the WL is different wrt the orbits, highlighting that the WL approximation is extremely close in practice.
>
> We will clarify this point further in the final version of the paper.
>
> [4] Babai, László, and Ludik Kucera. "Canonical labelling of graphs in linear average time." 20th annual symposium on foundations of computer science (sfcs 1979). IEEE, 1979.
> [5] Morris, Matthew, Bernardo Cuenca Grau, and Ian Horrocks. "Orbit-Equivariant Graph Neural Networks." The Twelfth International Conference on Learning Representations.
> [6] Chen, Guangyong, et al. "Alchemy: A quantum chemistry dataset for benchmarking ai models." arXiv preprint arXiv:1906.09427 (2019).
>
> ---
>
> **Q7: How sensitive are the results to the choice of α in the reliability check? Characterizing this would be helpful.**
>
> We tested the sensitivity of our results to the choice of the significance level in the reliability check. Specifically, we experimented with three commonly used values in statistical testing, i.e., $\{0.1,\ 0.05,\ 0.01\} $. The resulting performance on LR-EXP are reported below.
>
> As expected, all standard GNN models consistently fail to recognize any links, regardless of the significance level. Among the more expressive models, the majority show stable behavior across different values of significance, with fluctuations that remain within the standard deviation. BUDDY and NCN exhibit slightly larger fluctuations in absolute value, but the key point for this experiment is the expressiveness ranking across models. This ranking remains robust across all significance levels, with one minor exception: at significance level 0.1, Neo-GNN slightly surpasses NCN. However, this difference falls within the standard deviation range.
>
> |Model| 0.1|0.05|0.01|
> |--------|-----------|-----------|-----------|
> |GCN|0 ± 0|0 ± 0|0 ± 0|
> |GAT|0 ± 0|0 ± 0|0 ± 0|
> |GAE|0 ± 0|0 ± 0|0 ± 0|
> |SAGE|11 ± 0|8 ± 4|3 ± 0|
> | BUDDY|54 ± 5| 45 ± 1    | 59 ± 1    |
> | ELPH|61 ± 1|62 ± 7|62 ± 1|
> | NeoGNN | 76 ± 6|75 ± 2| 71 ± 2    |
> | NCN| 70 ± 1 |75 ± 1|81 ± 1 |
> | SEAL| 92 ± 0|97 ± 0| 97 ± 0 |
>
> We also note that the lack of a strictly monotonic trend in the number of reliably distinguished links when decreasing the significance level $\alpha$ is due to the **two-part structure** of our statistical test. Specifically, it combines:
>
> 1. The **Major Procedure**, which tests whether a model can distinguish between two *non-automorphic* links (i.e., tests if their representations differ significantly);
> 2. The **Reliability Check**, which ensures that the same model assigns *equal* representations to *automorphic* links (i.e., avoids false positives due to internal fluctuation).
>
> For each pair of non-automorphic links, we compute a threshold $\tau $, and accept the distinction only if the test statistic $T^2_{\text{major}} > \tau $ **and** the corresponding statistic for automorphic links $T^2_{\text{check}} < \tau $. This means that when $\alpha$ increases, some previously rejected distinctions may now pass the Major Procedure (leading to more links being distinguished), but at the same time, the Reliability Check may fail for others (causing fewer to pass the overall test). These two effects may balance out or interfere with each other, explaining why no simple monotonic pattern emerges in the results.

---

> > ### Comment · Reviewer_qEnu · 2025-08-01
> >
> > Thank you for your thorough response, and in particular for rerunning the experiments with GraphSAGE. I appreciate that the experiments confirm the theoretical results of Srinivasan et al. Indeed, I feel that the improvements that you've made make the submission stronger.
> >
> > An additional question -- and I'm legitimately unsure of the answer here -- is it possible to formalize the expressiveness of GraphSAGE with stochastic neighbor sampling within either your $k_\phi$ or $k_\rho$ functions? If so, it would be interesting to formalize the way that SNS improves the expected link-rep expressiveness. Additional questions include what happens in the limit that SNS includes all the neighbors (what you already have) and when SNS samples only one neighbor (a single random walk).

---

> > > ### Author Response · Authors · 2025-08-01
> > >
> > > We thank the reviewer for the thoughtful comment and the engaging follow-up.
> > >
> > > In models that adopt SNS, the embeddings become random variables rather than deterministic functions. In principle, it is possible to adapt our framework to such probabilistic methods by replacing embedding with a distribution over embeddings in all our definitions and theorems, for example by adopting the probabilistic notion of node embedding already formalized in Definition 12 of Srinivasan et al. [1]. Regarding this possibility, what would require further study— and represents an interesting direction for future work — is how to compare probabilistic and deterministic methods within a common evaluation framework. For example, while with SNS is it possible to distinguish structurally different links that standard GNNs fail to separate, it also introduces the possibility that structurally equivalent links receive different representations. This phenomenon is similar to what occurs in models that inject random features to increase expressiveness at the graph level [2], which also lie outside the classical WL hierarchy.
> > >
> > > [1] Srinivasan, B., & Ribeiro, B. (2020). On the Equivalence between Positional Node Embeddings and Structural Graph Representations. ICLR.
> > > [2] Sato, Ryoma. "Random Features Strengthen Graph Neural Networks." NeurIPS, 2020.

---

> > > > ### Comment · Reviewer_qEnu · 2025-08-01
> > > >
> > > > Thank you. Your reply makes sense. You've addressed all of my questions satisfactorily.

---

### Official Review · Reviewer_gepd · 2025-07-02

**Clarity:** 2
**Significance:** 3
**Originality:** 2
**Rating:** 4
**Confidence:** 3

**Summary:**

This paper studies how Graph Neural Networks (GNNs) represent links (pairs of nodes), which is less understood than node or graph-level tasks. The authors propose a new framework that helps describe and compare many existing link models. They build a hierarchy based on expressiveness and support it with both theory and experiments. They also design a small synthetic dataset (LR-EXP) to test whether models can tell different types of links apart, and analyze how symmetry in graphs affects performance.

**Questions:**

1. Can the framework be extended to non-MP models like GNN transformers?
2. What is the best trade-off between the number of GNN layers and neighborhood size?
3. Can you propose new methods or models that boost the existing performance based on the insight in this paper?

**Ethical Concerns:**

["NO or VERY MINOR ethics concerns only"]

**Final Justification:**

The author's rebuttal has addressed my concerns in terms of the paper's significance. They have also provided additional ablation and empirical studies. My previous proposed questions have been answered clearly. Therefore I would like to raise my score.

**Limitations:**

See above

**Quality:**

3

**Strengths And Weaknesses:**

### **Strengths**

- The proposed framework is simple and unifies many existing methods.
- The paper provides strong theoretical insights about when and why some models are more expressive.
- The LR-EXP dataset is designed to test link-level expressiveness and can be used for future studies.

### **Weaknesses**
- The paper is more like a benchmark paper instead of a research-track paper, as it mainly provides an evaluation method for existing ones.
- The datasets used in this study are limited in size.
- More ablation or empirical studies on design choices like GNN depth vs. neighborhood radius would help.

---

> ### Author Rebuttal · Authors · 2025-07-30
>
> **We would like to thank the reviewer for the thoughtful comments and useful suggestions.**
>
> ---
>
> **W1: The paper is more a benchmark paper of a research-track paper**
>
> While our paper introduces a new synthetic dataset and evaluates link prediction models on real-world benchmark, its primary contribution is theoretical. We propose a novel framework to formally analyze and compare the expressiveness of GNN-based link representation models. The paper presents new theoretical results (Theorems 3.2 and 3.3), establishes an expressiveness hierarchy of existing methods, and introduces a graph symmetry metric to connect theoretical insights with empirical performance. We believe, as also recognized by the other reviewers, that the strength and originality of these methodological and theoretical contributions make the paper well-suited for the Research track.
>
> ---
>
> **W2: The datasets used in this study are limited in size**
>
> We would like to clarify that our study includes both small and large-scale datasets. In particular, we evaluate on OGBL-Citation2 (2,927,963 nodes and 30,561,187 edges) and OGBL-PPA (576,289 nodes and 30,326,273 edges), which are among the largest publicly available link prediction benchmarks.
>
> That said, we emphasize that expressiveness is not inherently related to dataset size. Our goal is not to assess scalability, but to understand the expressiveness of different models.
>
> ---
>
> **W3: More ablation or empirical studies on design choices like GNN depth vs. neighborhood radius would help.**
>
> We thank the reviewer for this suggestion and have conducted additional ablation experiments to further analyze the effect of GNN depth and neighborhood radius on model expressiveness.
>
> In the original experiment reported in the paper, to ensure a fair comparison, we selected for each model the maximum neighborhood radius that makes it as expressive as possible while still respecting its design constraints. For example, in models like NCN, the radius is fixed by design and equal to 1, so it cannot be varied. In others, such as ELPH, the radius is tunable but is constrained to a maximum of 3 by the original authors for computational reasons, and we adopted that upper bound. As for GNN depth, we fixed the number of layers to 3 as commonly done in the literature [1].
>
> To explore this further, we conducted two additional experiments on the LR-EXP dataset:
>
> 1. **Fixed radius (1), varying the number of layers**
> 2. **Fixed number of layers (3), varying the neighborhood radius**
>
> The results on the LR-EXP dataset are reported below.
>
> **(1) Fixed Radius = 1**
>
> | # Layers | NCN     | NeoGNN     | ELPH     | SEAL     |
> |----------|---------|------------|----------|----------|
> | 1        | 74 ± 1  | 42 ± 4     | 40 ± 4   | 79 ± 3   |
> | 2        | 75 ± 1  | 42 ± 4     | 55 ± 4   | 89 ± 1   |
> | 3        | 75 ± 1  | 45 ± 3     | 62 ± 2   | 89 ± 2   |
>
> **(2) Fixed # Layers = 3**
>
> | Radius | NCN     | NeoGNN     | ELPH     | SEAL     |
> |--------|---------|------------|----------|----------|
> | 1      | 75 ± 1  | 45 ± 3     | 62 ± 2   | 89 ± 2   |
> | 2      | —       | 63 ± 3     | 61 ± 4   | 90 ± 3   |
> | 3      | —       | 75 ± 2     | 62 ± 7   | 97 ± 0   |
>
> > Note: For NCN, the radius is fixed by design to 1 and cannot be increased.
>
> These results are fully consistent with Theorem 3.2 (Point 2):
> - When fixing the radius, increasing the number of layers improves expressiveness in all the models.
> - When fixing the number of layers, increasing the radius also improves expressiveness in all the models.
>
> [1] Oono, Kenta, and Taiji Suzuki. Graph Neural Networks Exponentially Lose Expressive Power for Node Classification. ICLR 2020.
>
> ---
>
> **Q1: Can the framework be extended to non-MP models like GNN transformers?**
>
> The current version of the framework is specifically designed to model message-passing based link representation methods, and does not extend to non-MP architectures such as GNN transformers.
>
> This focus is intentional: MP models are the foundation of most existing theoretical work in graph learning, and their expressiveness in the context of link representation is still not fully understood. Just as MP GNNs served as the basis for early expressiveness studies in node and graph classification, later extended to more complex architectures like transformers, we believe that starting from MP models is a necessary and meaningful first step.
>
> As stated in the Limitations section (Section 6), extending the framework to account for transformer-based or attention-based architectures is a valuable direction for future research, and we thank the reviewer for highlighting this point.
>
> ---
>
> **Q2: What is the best trade-off between the number of GNN layers and neighborhood size?**
>
> We would like to clarify that we do not claim the existence of a trade-off between the number of GNN layers and the neighborhood radius. In fact, according to Theorem 3.2, increasing **both** the number of layers and the neighborhood radius leads to higher expressiveness for message-passing models in link representation.
>
> If a trade-off exists, it is not between these two parameters, but rather between **expressiveness and computational cost**. Our framework allows us to reason about this trade-off more precisely. Specifically, it suggests that:
>
> 1. There is no need to go beyond 1-WL expressivity in the base GNN, as increasing base expressiveness adds cost without increasing expressiveness in pair-wise representation (Theorem 3.2 point 1).
> 2. It is more efficient to use node representations computed with message passing layers, in a smaller neighborhood, rather than using weaker node features (like simple structural counts) over a larger neighborhood (Theorem 3.2 point 2).
>
> ---
>
> **Q3: Can you propose new methods based on the insight in this paper?**
>
> Theorem 3.2 provides insights for designing expressive and efficient link prediction models, specifically, by using node representations computed via 1-WL message passing over a limited-radius neighborhood and a suitable number of layers.
>
> Additionally, the symmetry metric helps guide **model selection**: on datasets with low structural symmetry, simple pure GNNs already suffice, avoiding unnecessary complexity and low generalization ability. On highly symmetric datasets, more expressive models are worth the added computational cost, as they can significantly improve performance.

---

> > ### Comment · Reviewer_gepd · 2025-08-08
> >
> > Thanks for the author's rebuttal. I have read the other reviewers' replies and agree that most of the concerns have been addressed. Therefore, I would like to raise my score. Thanks for your effort!

---

### Official Review · Reviewer_eEem · 2025-07-02

**Clarity:** 3
**Significance:** 2
**Originality:** 2
**Rating:** 4
**Confidence:** 3

**Summary:**

This paper studies the expressiveness of GNNs for link prediction and presents a new unifying framework that formally characterizes existing link prediction models. The authors also present a new synthetic dataset to specifically evaluate link-level expressiveness and highlight the role of graph symmetry in the performance of GNN models for link prediction.

**Questions:**

See weaknesses. Additionally:
- In the real-world benchmark analysis, can the authors comment on why the more expressive models (BUDDY, SEAL, etc.), perform so much worse in lower-symmetry settings?
- Do the authors have any intuition as to how their framework relates to knowledge graph embedding (KGE) models, which also focus on link prediction?
- Could the authors please clarify if they are using the oblivious or non-oblivious (i.e. folklore) version of the WL hierarchy (see Grohe, 2021 for details)? For completeness, it would be useful to explicitly state this.

References:

M. Grohe. The logic of graph neural networks. In _LICS_, 2021.

**Ethical Concerns:**

["NO or VERY MINOR ethics concerns only"]

**Final Justification:**

While the general framework is interesting, the overall scope of the work is somewhat limited, since the analysis excludes higher-order GNNs. However, the framework's ability to accomodate several current methods is technicall solid, and thus I maintain my score.

**Limitations:**

Yes

**Paper Formatting Concerns:**

I have no formatting concerns.

**Quality:**

3

**Strengths And Weaknesses:**

Strengths:
- The paper is well written and easy to follow
- The proposed framework is clear and appears naturally for comparing these models
- Experimental results on the newly presented synthetic dataset support the theoretical expressivity results
- Analysis on real-world performance and the effect of graph symmetry provides interesting insights

Weaknesses:
- The scope of the proposed framework is not fully clear within the context of existing literature. How does it compare to works such as Huang et al., 2023, which studies the expressivity of GNNs for link prediction over heterogeneous graphs or Zhang et al., 2021, which presents theoretical results on multi-node representation learning?
- Motivation-wise, the usefulness of this new characterization is still a bit unclear to me due to its limited scope. Link representations are already addressed in the traditional WL hierarchy by higher order GNNs which can generate node-pair representations for link-level tasks.

References:

Huang et al. A Theory of Link Prediction via Relational Weisfeiler-Leman on Knowledge Graphs. In _NeurIPS_, 2023.

Zhang et al. Labeling Trick: A Theory of Using Graph Neural Networks for Multi-Node Representation Learning. In _NeurIPS_, 2021.

---

> ### Author Rebuttal · Authors · 2025-07-30
>
> **We would like to thank the reviewer for the thoughtful comments and useful suggestions.**
>
>
> ---
>
>
>
> **W1: Comparison with Zhang et al., 2021 and Huang et al., 2023**
>
> Zhang et al. (2021) propose the *labeling trick*, a specific method designed to improve the expressiveness of GNNs for link prediction. In our work, we present a general framework that *subsumes* all message-passing methods for link representation, including the labeling trick (see the last row of Table 1). The role of the framework is to provide a formal lens through which different link representation models can be expressed using the same building blocks, characterized by the functions $\phi $, $\rho$, and the parameters $ l $ and $m $. By rewriting each method within the framework, we can precisely understand its expressiveness and compare it to others in a principled way. For instance, thanks to Theorem 3.2, we can conclude that the labeling trick is more expressive than any other existing method based on structural features. We will better explain it in the revised version of the manuscript.
>
> Higher-order GNNs could, in principle, be used  to construct expressive link representations. We note, however, that the quadratic complexity of “proper” order-2 GNNs has prevented practitioners from actually following that approach. Thus, the approach by Huang et al. (2023), though inspired by higher-order message passing, effectively still computes node level representations (but relative to a second, fixed reference node). The goal of our paper was to provide a framework that accommodates a large number of current methods that have been applied in practice. Extending the framework  to include also (proper) higher-order approaches would be interesting from a purely theoretical point of view, but is outside our current scope.
>
> ---
>
> **W2: Usefulness of the framework with respect to WL hierarchy**
>
> As noted in our response to W1, while it is true that higher-order GNNs (e.g., 2-WL-based architectures) could, in principle, produce expressive node-pair representations, their quadratic complexity has significantly limited their practical adoption. Many alternative link prediction models have been proposed that rely on other strategies, such as structural features or neighborhood-based heuristics, to achieve higher expressiveness in a more computationally efficient way. The goal of our paper is to provide a unified framework that encompasses all these methods (and potentially also future ones), and to offer theoretical tools to compare their expressiveness in a principled manner.
>
> ---
>
> **Q1: Why do the more expressive models perform so much worse in lower symmetry setting?**
>
> More expressive models are explicitly designed to distinguish non-automorphic links with automorphic endpoints (i.e., dashed red links in Figure 2). However, in low-symmetry graphs, such ambiguous links are rare. These graphs tend to have highly informative features and irregular structures, making most links trivially distinguishable from their endpoints alone. As a result, standard GNNs are often sufficient to achieve strong performance.
> In this context, the added expressiveness of models like SEAL and BUDDY does not yield a clear advantage, but it does not systematically hurt either. For instance, on the four real-world datasets with symmetry below 0.1, BUDDY performs better than GCN in two cases and worse in two; SEAL outperforms GCN in one case. This suggests that in low-symmetry settings, simpler models are often sufficient and the performance gap narrows.
>
> ---
>
> **Q2: Do the authors have intuitions on the connection with KGE?**
>
> Our framework could be quite easily generalized to directed and multi-relational graphs, including knowledge graphs. We did not go in this direction for the time being, because, on the one hand, this would cause a significant blowup in complexity of notation and the statement of Theorem 3.2. On the other hand, such structurally richer graphs would typically exhibit much lower symmetry levels, thereby alleviating the limitations of node-level representations for link prediction. This is particularly true for knowledge graphs. Indeed, for link prediction in the transductive setting (in particular: knowledge graph completion) it would be admissible to include unique node identifiers as node attributes, in which case our expressivity problem disappears completely. This also is a reason why our synthetic data focuses on inductive link prediction on many small graphs.
> As for KGE models, positioning them within our framework is less straightforward, since many do not rely on message passing and operate under fundamentally different assumptions. A detailed investigation of how KGE approaches relate to our framework is an interesting open question, and we consider this a valuable direction for future work.
>
> ---
>
> **Q3: Which version of the WL-hierarchy are the authors using?**
>
> We thank the reviewer for the opportunity to clarify this important point. We use the oblivious version of the Weisfeiler–Leman test. We will make this explicit in the preliminaries section of the paper to avoid any ambiguity.

---

> > ### Comment · Reviewer_eEem · 2025-08-04
> >
> > I thank the authors for their detailed response - they have addressed my questions, and I am happy to keep my score.

---

### Note · Authors · 2025-08-12

We thank all reviewers for their thoughtful feedback and constructive engagement throughout the rebuttal and discussion phases. Following our rebuttal and additional experiments, the only reviewer who initially gave a score of 3 explicitly raised their rating, acknowledging that their main concerns had been addressed. All other reviewers confirmed their positive evaluations.

Our key contributions are:

1) A unifying theoretical framework that formally subsumes all SOTA MP-based link representation methods.

2) New theoretical results (Thm. 3.2 & 3.3) establishing a principled hierarchy of link-level expressiveness, connecting depth, neighborhood radius, and the expressive power of the base node-level MP.

3) A synthetic evaluation protocol for assessing the expressive power of link-level MP methods, including the first synthetic benchmark explicitly designed for this purpose, together with a dedicated evaluation framework.

4) An empirical investigation of the practical relevance of expressiveness in link representation: we introduce a graph symmetry metric to quantify structural ambiguity among links, showing that while simple models suffice in low-symmetry settings, more expressive models significantly outperform them as symmetry increases.

5) Guidance for model design and selection: our theoretical results (points 1 and 2) provide actionable principles for designing future models that balance expressiveness and computational cost, while our empirical findings (point 4) inform when using more expressive (and costly) models is truly beneficial in practice.

In response to reviewer questions, we:

1) Clarified positioning w.r.t. Zhang et al. 2021, Huang et al. 2023, and Srinivasan et al. 2020, showing how our framework subsumes prior methods and extends beyond case-by-case analyses.

2) Added ablation studies on depth vs. neighborhood radius, confirming Thm. 3.2’s predictions.

3) Extended the analysis to GraphSAGE with stochastic neighbor sampling, aligning with the theoretical expectations of Srinivasan et al. 2020.

4) Detailed computational costs, identifying NCN/BUDDY as the best expressiveness–cost trade-offs.

5) Explained symmetry metric computation and its real-world implications.

Reviewers recognized the value of our theoretical and methodological contributions, and no major concerns were raised during the discussion phase.

We thank the AC and SAC for their work in overseeing the review process.

---

### Decision · Program_Chairs · 2025-09-17

**Decision:**

Accept (spotlight)

**Comment:**

This paper makes a substantive contribution by shifting expressiveness analysis to link representations, unifying mainstream message-passing link models, deriving a clear expressiveness hierarchy, and providing actionable design guidance. Theoretical results are precise, and the LR-EXP suite plus symmetry metric effectively link theory to practice. Reviewers praised clarity and impact; rebuttal experiments addressed concerns on scope, ablations, and empirical grounding, leading to score increases. Remaining issues are minor and easily fixed. Overall, this is a clear, well-supported, and impactful contribution deserving strong acceptance.